# Development and validation of reliable astaxanthin quantification from natural sources

**Inga K. Koopmann**[ID]**, Annemarie Kramer, Antje Labes**[ID]*

ZAiT, Center for Analytics in Technology Transfer of Bio and Food Technology Innovations, Flensburg University of Applied Sciences, Flensburg, Schleswig-Holstein, Germany

* antje.labes@hs-flensburg.de

**Data Availability Statement:** All relevant data are within the paper and its Supporting information files.

## Abstract

Astaxanthin derived from natural sources occurs in the form of various esters and stereomers, which complicates its quantitative and qualitative analysis. To simplify and standardize astaxanthin measurement with high precision, an enzymolysis-based astaxanthin quantification method was developed to hydrolyze astaxanthin esters and determine free astaxanthin in all its diastereomeric forms. Astaxanthin standards and differently processed *Haematococcus pluvialis* biomass were investigated. Linear correlation of standards of all-*E*-astaxanthin was observed in a measurement range between extract concentrations of 1.0 µg/mL and 11.2 µg/mL with a coefficient of variation below 5%. The diastereomers 9*Z*-, and 13*Z*-astaxanthin, and two di-*Z*-forms were detected. In contrast to the measurement of standards, the observed measurement range was extended to 30 µg/mL in extracts from *H. pluvialis*. The nature of the sample had to be taken into account for measurement, as cell, respectively, sample composition altered the optimal concentration for astaxanthin determination. The measurement precision of all-*E*-astaxanthin quantification in dried *H. pluvialis* biomass (1.2–1.8 mg dried biomass per sample) was calculated with a coefficient of variation of maximum 1.1%, whereas it was below 10% regarding the diastereomers. Complete enzymolysis was performed with 1.0 to 2.0 units of cholesterol esterase in the presence of various solvents with up to 2.0 mg biomass (dry weight). The method was compared with other astaxanthin determination approaches in which astaxanthin is converted to acetone in a further step before measurement. The developed method resulted in a higher total astaxanthin recovery but lower selectivity of the diastereomers. The reliability of photometric astaxanthin estimations was assessed by comparing them with the developed chromatographic method. At later stages in the cell cycle of *H. pluvialis*, all methods yielded similar results (down to 0.1% deviation), but photometry lost precision at earlier stages (up to 31.5% deviation). To optimize sample storage, the shelf life of astaxanthin-containing samples was investigated. Temperatures below -20°C, excluding oxygen, and storing intact *H. pluvialis* cells instead of dried or disrupted biomass reduced astaxanthin degradation.

**Funding:** IKK, AK, AL: The ZAiT is part of the project Grenzland INNOVATIV Schleswig-Holstein [innovative border region Schleswig-Holstein] funded by the Ger-man Federal Ministry of Education and Research (BMBF) in context of "Inno-vative Hochschule" [innovative university], https://www.bmbf.de/bmbf/en/home/home_node.html. We acknowledge financial support by Land Schleswig-Holstein (federal state Schleswig-Holstein) within the funding program Open Access-Publikationsfonds. The funders had no role in study design, data collection and analysis, decision to publish, or preparation of the manuscript.

**Competing interests:** IK has worked for Sea and Sun Technology, a provider of H. pluvialis biomass for the study. All authors have declared that no other competing interests exist. This does not alter our adherence to PLOS ONE policies on sharing data and materials.

## Introduction

Astaxanthin (3,3′-dihydroxy-β,β′-carotene-4,4′-dione) is a secondary ketocarotenoid. It has a hydrocarbon backbone that comprises a central, delocalized π-electron system. β-ionone rings terminate the hydrocarbon chain at both ends [1, 2]. The presence of one hydroxy- and one oxo-group at each of these terminal rings further classifies it as xanthophyll. Its biosynthesis has been observed mainly in microalgae [3–12] but also in a few protists [13–15], bacteria [16–20], archaea [21] as well as yeasts [22, 23], and very few plant species [24, 25]. Astaxanthin exhibits properties against photooxidative stress by scavenging free radicals [26–31]. These antioxidative effects were also observed in mammalian cells *in vitro* and *in vivo* [32–37] and led to its application as a nutritional supplement, food and feed additive, and in cosmetics. Astaxanthin from natural sources has been authorized for human consumption in many countries worldwide [38]. Few minor adverse effects of dosed astaxanthin consumption were observed [38]. Allergic reactions to the consumption of *H. pluvialis* proteins cannot be excluded, but their likelihood has been considered low in tested astaxanthin-rich novel food ingredients by the European Food Safety Authority (EFSA) [39]. It can also be produced synthetically, but its natural form has gained interest with respect to consumer demands [40, 41]. A major source for the biotechnological production of astaxanthin is the green alga *Haematococcus pluvialis* [11, 42]. It contains 1.9 to 7.0% astaxanthin of its dry weight in certain life cycle stages and under stress conditions [43–48]. As in other producing organisms, astaxanthin in *H. pluvialis* is derivatized at one or both hydroxyl groups with various fatty acids, mainly leading to the formation of monoesters (76–94%), besides diesters (2–23%) and free astaxanthin (0.3–4%) [49–53]. The most common fatty acids occurring in these combinations are C18:1 and C18:2, amongst many others [49–52]. The conjugated double-bound system can theoretically form a variety of diastereomers [54]. The most abundant form in *H. pluvialis* is the all-*E* configuration followed by the sterically unhindered 9*Z*-, 13*Z*- and 15*Z*-forms [55–59] as well as corresponding di-*Z*-forms [50, 58, 60]. Due to the two stereogenic carbon atoms C-3 and C-3', three or four enantiomers are possible when a symmetrical or asymmetrical *Z*-form is considered, respectively (3*S*,3'*S*, 3*R*,3'*R*, 3*S*,3'*R* / 3*R*,3'*S*). In *H. pluvialis*, the ratio of the astaxanthin enantiomers was reported to be 88:10:2 and 99:0:1 (3*S*,3'*S*, 3*R*,3'*R* and 3*S*,3'*R*) [57, 61, 62].

Determining total astaxanthin in biological samples is crucial on the analytical and production scale. Most studies use estimating, photometric quantification methods instead of precise determination of total astaxanthin including its isomers for the means of hyphenated chromatography. Photometry enables simple and fast determination without the need for sophisticated equipment [23, 43, 63–65]. The biological variability of astaxanthin with all its different isomeric forms and esters [49, 51] complicates the exact quantification of astaxanthin. Moreover, astaxanthin is sensitive to treatment with solvents [54, 66, 67]. Access to sophisticated hyphenated chromatography protocols is limited. Here we demonstrate a fast, robust, and reliable method to get reproducible and comparable results of an overall astaxanthin content.

Many available methods to determine the astaxanthin content in *H. pluvialis* have been described. The most sophisticated ones extract, identify, and quantify a variety of esters and isomers with liquid chromatographic and spectrophotometric/mass spectrometric methods [46, 49–51, 64, 68–74]. However, this is time-consuming, as liquid chromatographic methods that differentiate the varying esters require much time for separation. Correct evaluation is elaborate because absorption spectra of carotenoids are similar. Moreover, correct quantification is difficult due to the different absorption coefficients of the geometric isomers.

A faster and more general approach is to extract all carotenoids and determine astaxanthin photometrically by estimating its proportion of total carotenoids [23, 43, 63]. The methods optionally include or destroy residual chlorophylls with a chemical treatment before measurement [45, 64, 65, 75, 76]. Recent approaches suggested the use of a higher wavelength (530 nm) than the absorbance maximum of astaxanthin for quantification to avoid overlapping absorption by chlorophylls and other carotenoids [48, 59, 77]. These photometric methods are cost-extensive, though the estimation may lead to unpredictable deviations, and the chlorophyll destruction may also impact the carotenoids. The geometrical isomers of astaxanthin have similar absorption spectra [50, 58–60, 66, 78], though different extinction coefficients [79], which impedes their individual quantification. To refine those photometric methods, astaxanthin can be fractionated by chromatographic methods (thin layer or column chromatography) beforehand and the corresponding fractions can be measured photometrically, applying the respective absorption coefficients [80–82]. However, the correct merging of the many esters and isomers is still difficult, leading to inaccuracy and deviation.

An option is to saponify the esters and isolate and quantify the resulting free astaxanthin with a shorter chromatographic method. Here, different methods for deesterification have been applied. Alkaline treatments with NaOH [55, 70, 83–88] or KOH [86, 89] are possible but can cause changes in the structural conformation and/or degradation of carotenoids [77, 85, 86, 90]. Thus, another approach is enzymolysis with esterases [61, 83, 85, 91, 92], lipases [93–95] or whole-cell catalysts [96].

The enzymolysis of astaxanthin is a promising compromise to quantify astaxanthin precisely while minimizing time consumption. Based on the enzymolysis of carotenoid esters, developed by Jacobs et al. [91], methods for quantification of astaxanthin were adapted by various studies and protocols [61, 83, 87, 92, 97–99]. However, to our knowledge, an in-depth evaluation of the process boundaries, detection limits, accuracy, and applicability of various astaxanthin-containing natural sources has not yet been performed. Therefore, a method for robust and reliable determination of astaxanthin from *H. pluvialis* was developed and validated, which was still faster than methods determining the various esters of astaxanthin. The five step-method comprises the preparation of biological samples and astaxanthin extraction, enzymolysis, liquid-liquid extraction of astaxanthin, processing of the resulting extract, and measurement with ultra-high performance liquid chromatography (UHPLC) and UV/VIS spectrometry. Particular emphasis was set to relevant factors: Enzyme and biomass amount, shelf life, incubation time, quantification limits, linearity, precision, systematic errors, robustness, and isomerization effects were tested. The final method was compared to photometric astaxanthin determination approaches. We aimed to develop a method that is applicable at all different stages of astaxanthin production and works with different formulations of astaxanthin. Therefore, it was tested with commercially available *H. pluvialis* and astaxanthin samples.

## Materials and methods

### Chemicals and reagents

Analytical grade acetone (SupraSolv), petroleum ether, and acetonitrile (hypergrade) were obtained from Merck (Darmstadt, Germany). Ethanol and Tris(hydroxymethyl)aminomethan (TRIS) (≥ 99.9%) were provided by Carl Roth (Karlsruhe, Germany) and formic acid (99% ULC/MS) by Biosolve (Valkenswaard, Netherlands). Hydrochloric acid for pH value adjustment was purchased from Merck (Darmstadt, Germany). Cholesterol esterase from *Pseudomonas* sp. for enzymolysis was obtained from MP Biomedicals (Eschwege, Germany). All-*E*-astaxanthin standard in its free form (SML0982, ≥ 97%, 3S,3'S, from *Blakeslea trispora*) was

provided by Sigma-Aldrich (Taufkirchen, Germany) and astaxanthin monopalmitate (1017, 3RS,3'RS) by CaroteNature (Münsingen, Swiss).

## H. pluvialis biomass and astaxanthin containing extracts

Various commercially available sources of *Haematococcus pluvialis* biomass containing astaxanthin were used: Dried and disrupted cysts were obtained from Golden Peanut (Garstedt, Germany). Sea & Sun Technology GmbH (Trappenkamp, Germany) provided concentrated *H. pluvialis* aplanospores from seven different batches in nutrient depleted media. Sea & Sun Technology also provided oleoresins of *H. pluvialis* obtained from supercritical $CO_2$ (SC-$CO_2$) extraction either pure or diluted in ethanol. Cultivation and harvest parameters are not available on those samples, but for two batches of concentrated, fresh biomass obtained from Sea & Sun Technology. Those were taken from day 22 to 28 of a light and nutrient-induced stress phase. All had been cultivated in BG11 medium (Culture Collection of Algae at the University of Göttingen): 17.6 mM $NaNO_3$, 0.18 mM $K_2HPO_4$ x $3H_2O$, 0.3 mM $MgSO_4$ x $7H_2O$, 0.25 mM $CaCl_2$ x $2H_2O$, 0.031 mM citric acid, 0.023 mM ferric ammonium citrate, 0.003 mM $Na_2EDTA$ x $H_2O$, 0.19 mM $NaCO_3$ and 1 mL/L trace metal solution made of 1.0 mM $H_3BO_3$, 1.0 mM $MnSO_4$ x $H_2O$, 1.00 mM $ZnSO_4$ x $7H_2O$, 0.01 mM $(NH_4)_6Mo_7O_{24}$ x $4H_2O$ and 0.1 mM $CuSO_4$ x $5H_2O$. These were used exclusively for the experiments in which the developed method was to be compared with photometric methods.

## Disruption of H. pluvialis and astaxanthin extraction

Undisrupted biomass was bead-milled to ensure astaxanthin accessibility during further processing. Therefore, 0.2–4.0 mg cell dry mass were weighed into lysis tubes type C (ceramic beads, diameter 0.4–0.6 mm) by Analytik Jena (Jena, Germany). 500 μL acetone were added for astaxanthin extraction. When analyzing aplanospores in liquid medium, volumes of equal to or less than 300 μL of well-homogenized samples were filled into the same lysis tubes and matched to a final amount of 0.5–2.0 mg of biomass. The tubes were filled up to 500 μL with acetone accordingly. The cysts were broken mechanically in a swing mill (MM 2000, Retsch, Haan, Germany) at 27 Hz for 3 minutes without cooling and centrifuged at 10,000 x *g*. The supernatant of both sample types was transferred to a centrifuge tube, and 500 μL of fresh acetone were added to the lysis tube. This procedure was repeated twice until the supernatant and the residual biomass were colorless.

## Astaxanthin deesterification by enzymolysis

The preparation for enzymolysis varied with the sample type. Using concentrated fresh or dried biomass, the combined supernatants obtained from astaxanthin extraction were filled up to 3 mL with acetone ($\triangleq$ 90–100% v/v). Highly viscous oleoresins were weighed directly into centrifuge tubes with 0.4 to 15.8 mg and diluted in 3 mL acetone. Oleoresins dissolved in ethanol were used at 0.1 to 3 mL and filled up with acetone to 3 mL. 2 mL of 50 mM TRIS buffer (pH 7 at 21°C) and 600 μL cholesterol esterase solution at a concentration of 3.3 U/mL suspended in the same buffer were added. Accordingly, final cholesterol esterase concentration in the sample was 2.0 units or 0.36 units per mL. The tubes were incubated at 37°C in a water bath and mixed gently every 10 minutes. Astaxanthin was recovered by liquid-liquid extraction with 2 mL ($\triangleq$ 26% v/v) of petroleum ether. The mixture was shaken vigorously for 10 s to ameliorate astaxanthin transfer into the petroleum ether. Subsequent phase separation was enhanced by centrifugation at 3,000 x *g* for 1 minute. The upper, astaxanthin-containing phase

was filtered (0.45 μm, PET) into amber vials. It was either measured directly or stored at -21˚C for one night before analysis.

## Analysis and quantification of free astaxanthin and lutein by UHPLC-PDA-MS

Extracts were vortexed (Vortex 3, IKA-Werke GmbH und Co. KG, Staufen, Germany) and ultrasonicated for 20–30 s (Sonorex Super RK 103 H, Bandelin electronic GmbH und Co. KG, Berlin, Germany) before analysis if they had been stored overnight. Qualification and quantification of astaxanthin were performed by UHPLC using an ACQUITY Arc system by Waters (Milford, MA, USA) equipped with a sample manager (FTN-R), a quaternary solvent manager (R), an UV/Vis detector (2998 PDA Detector), and a mass spectrometer (Acquity QDa Detector). A C18-column (Cortecs C18 2.7 μm, 90 Å, 3.0 x 100 mm) was operated at 40˚C. The injection volume was 5 μL. Starting conditions were 70% millipore water and 30% acetonitrile, both containing 0.1% formic acid. Over four minutes, the gradient increased linearly to 90% acetonitrile and 10% millipore water. This ratio was kept isocratic for five minutes. A rinsing step on 100% acetonitrile was attached for a further 2.5 minutes. 3.5 minutes were used for regeneration of the starting conditions. Flow velocity was 0.5 mL/min. Optical spectra were measured in a range of 200 to 800 nm, and astaxanthin data were analyzed and quantified at its determined absorbance maximum of 474 nm. The absorption maximum of lutein was shown to be at 448 nm. This wavelength was used for quantification. The mass spectrometer with electrospray ionization (ESI) was operated in positive mode with a cone voltage of 15 V and a probe temperature of 600˚C, measuring in a range of $m/z$ 150 to 1250. For further accuracy, the mass of astaxanthin was observed by selected ion recording (SIR) at $m/z$ 598 $[M+H]^+$ and lutein at $m/z$ 570 $[M+H]^+$. Besides the major all-*E*-astaxanthin peak, peaks with UV/Vis absorption spectra corresponding to *Z*-astaxanthin isomers [50, 58–60, 66, 78] that were additionally accompanied by peaks with the mass of astaxanthin in SIR were assigned to 9*Z*- and 13*Z*-astaxanthin and several di-*Z*-isomers. However, as the latter are difficult to differentiate without further validation, they were summed and are consecutively termed di-*Z*-isomers. Quantities of all diastereomers were estimated by using the quantification of all-*E*-astaxanthin, corrected by factors adjusting the different extinction coefficients published by Bjerkeng et al. [79], namely 1.20 for 9*Z*-astaxanthin, 1.56 for 13*Z*-astaxanthin, and 1.11 for the di-*Z*-isomers.

## Calibration curve—Astaxanthin

For identification and quantification, free all-*E*-astaxanthin was used at concentrations of 0.5–45.0 μg/mL in acetone. Blank acetone was applied for zero value determination. Samples of 0.5 to 54.8 μg free astaxanthin and 1.0 to 62.6 μg astaxanthin monopalmitate were subjected to the deesterification process to determine their recoveries. Both were diluted in acetone and treated as described above. In the processing of free astaxanthin, the cholesterol esterase solution was replaced with the same amount of TRIS buffer. For quantification of monopalmitate-ester derived free astaxanthin, a proportion of 69.4% w/w astaxanthin was assumed by molecular weight calculation.

## Calibration curve—Lutein

To identify and quantify lutein from *H. pluvialis*, free lutein standard in concentrations of 1.0 to 61.5 μg/mL dissolved in acetone was used for calibration. Linear regression was performed by ordinary least squares and forced through zero for optimal approximation to standards recovered in petroleum ether.

## Optimization of enzymolysis and liquid-liquid extraction

**Enzyme amount and duration of enzymolysis.** The amount of cholesterol esterase was varied between 0.05 to 2.0 units per reaction for equal amounts of astaxanthin esters in 0.4 mg *H. pluvialis* powder (Golden Peanut) to pinpoint optimal enzyme concentration for astaxanthin conversion during 0.75 hours of incubation. In addition, the incubation time of enzymolysis was doubled to 1.5 hours using 0.5 units and 2.0 units of cholesterol esterase.

**Influence of ethanol on astaxanthin enzymolysis.** The recovery of astaxanthin in samples processed in the presence and absence of ethanol was compared. Therefore, two different ethanolic SC-CO$_2$ extracts (n = 7 and n = 5) of *H. pluvialis* with volumes between 150 and 1000 μL were used. They were either fed directly to the deesterification process (n = 5 and n = 3) or ethanol was evaporated at 40˚C under nitrogen, and the sample was subsequently resuspended in acetone and TRIS buffer containing 2.0 units of cholesterol esterase for deesterification (n = 2 and n = 2). All samples were enzymolyzed, extracted, and astaxanthin content was quantified based on UHPLC-PDA measurements as described above.

**Influence of solvents used in liquid-liquid extraction on astaxanthin quantification.** The volume of the upper petroleum ether layer was determined after liquid-liquid extraction to determine extract concentrations. Therefore, blank samples (n = 8) consisting of 3 mL acetone and 2.6 mL TRIS buffer were heated to 37˚C for 20 minutes in a water bath. Afterward, the samples were shaken vigorously with 2 mL petroleum ether, centrifuged at 3,000 x *g* for 1 minute, and cooled to 21˚C in a water bath. The complete upper layer was transferred into reaction tubes and weighed (n = 4). Its density was determined by measuring the upper layer of the remaining samples with a pycnometer (n = 4). The corresponding volume was calculated.

To quantify the influence of ethanol on the volume of the upper extraction phase, various volumes of ethanol (0 to 2010 μL) were filled up to 3 mL with acetone and treated similarly to the previous experiment.

## Extract processing

After liquid-liquid extraction, various processing procedures of the petroleum ether fraction were compared to reach the maximum recovery of astaxanthin. Therefore, 20 μg of free astaxanthin standard were deesterified and extracted with petroleum ether as described above. The upper layer was treated in three ways: (1) The extraction phase was not treated. (2) An aliquot of the extraction phase was dried under nitrogen and redissolved in the same volume of acetone. (3) Most of the extraction phase was transferred to another tube, and fresh petroleum ether was added to the original sample. The extraction and transfer steps were repeated twice until the upper layer was colorless. The combined fractions were dried at 40˚C under nitrogen and redissolved in 2 mL of acetone. The samples were filtered, and astaxanthin was measured by the UHPLC-PDA-MS method. All experiments were performed in triplets or quartets.

## Evaluation of detection limits and linearity of astaxanthin determination

Detection limits and linearity of astaxanthin measurement were determined by varying the amount of *H. pluvialis* powder (Golden Peanut) from 0.04 mg to 2.0 mg in triplets. 3.3 and 4.0 mg were tested without replicates. They were enzymolyzed with a constant concentration of 2.0 units cholesterol esterase in 3 mL of acetone and 2.6 mL of TRIS buffer and 0.75 hours of incubation.

## Determination of measurement precision

The astaxanthin content of two samples from different packages of dried *H. pluvialis* biomass (Golden Peanut) was measured using the standard enzymolytic UHPLC-PDA procedure described above. Measurements were performed once on three and four different days to determine the variation between experiments carried out independently. Between 1.2 and 1.8 mg of sample were applied.

## Method adjustment

**Liquid cultures and oleoresins.**   Various liquid culture batches of *H. pluvialis* biomass (Sea & Sun Technology) concentrated at 7.1, 39.5, 182, and 262 g/L were examined. For disruption, volumes of 1 to 45 µL of these samples were transferred to lysis tubes to get final amounts of 0.1 to 2.0 mg of biomass. Further processing was accomplished as described above.

Additionally, four oleoresins of *H. pluvialis* extracted with SC-$CO_2$ were investigated. Therefore, 0.15 to 1.8 mg of oleoresin were directly added to the deesterification solution. Enzymolysis was carried out with 2.0 units and 4.0 units of cholesterol esterase per reaction. Extraction and quantification were performed as described above.

**Sample mixtures.**   Various astaxanthin-containing samples were deesterified individually and merged to examine cross-interactions. The individual samples contained either 240 µg of *H. pluvialis* powder (Golden Peanut), 16 µg of astaxanthin monopalmitate, or 12 µg of free astaxanthin. Maximum 4.5% w/w of the *H. pluvialis* powder were considered to be free astaxanthin by measurements. Astaxanthin monopalmitate consists of about 69.4% w/w of free astaxanthin. Thus, the final extracts of the samples contained 4.2 to 4.7 µg/mL of free astaxanthin. Moreover, three samples were prepared, two containing mixtures of two of the single solutions, and one sample containing all three. In these, the individual samples were represented at identical amounts as in the samples containing only one of the analytes.

After enzymolysis and recovery in petroleum ether, three aliquots of 600 µL were taken from each sample. They were prepared either with 63 µL pure acetone, 31.5 µL pure acetone and 31.5 µL of astaxanthin standard solution containing 3.6 µg free astaxanthin, or with 63 µL of astaxanthin standard solution containing 7.2 µg free astaxanthin. Furthermore, 600 µL pure petroleum ether were enriched with acetone and astaxanthin like the samples. The experiment was performed once.

## Method comparison to photometric astaxanthin estimation

To compare the developed method to simpler photometric approaches, the astaxanthin content of several *H. pluvialis* samples (Sea & Sun Technology) taken towards the end of a cultivation phase were measured with UHPLC-PDA-MS and photometrically. From two individual culture batches, samples were taken on days 22–24 and 27–28 of the cultivation. For each UHPLC measurement, 1.0 mg of *H. pluvialis* biomass was used. Extraction, enzymolysis, and quantification were carried out as described above. All samples were measured once. For photometric measurements, samples were extracted similarly. The extracts were measured at λ = 470 nm and with a wavelength scan from 300 to 700 nm in steps of 2 nm. Astaxanthin content was calculated using two different approaches: (1) Lambert-Beer was applied with the concentration of astaxanthin ($C_{ax}$) in g/mL, the optical path length d in cm, the absorption at the wavelength represented with the subscript number and the 1%-absorption coefficient $A_{1cm}^{1\%}$ $\varepsilon_{ax}$ =1980 [100mL/(g$^*$cm)] of a carotenoid mixture in 80% acetone and 20% water [100]. A factor of 0.8 for the astaxanthin proportion of total carotenoids was assumed based on various literature data [45, 53, 59, 101]. This (Eq 1) will be termed "general equation" hereafter. (2)

The calculations of Lichtenthaler [100] for chlorophylls and total carotenoids were used to determine the carotenoid fraction. See Eqs 2.1–2.3 with the concentration of chlorophyll $a$ ($C_a$), chlorophyll $b$ ($C_b$), and of total carotenoids ($C_{x+c}$) in μg/mL in acetone with 20% water, and the absorptions at corresponding wavelengths represented with the subscript numbers. Again, the astaxanthin proportion was calculated by multiplying the total carotenoid content with a factor of 0.8 (Eq 2.4).

$$C_{ax} = \frac{0.8\, A_{470}}{\varepsilon_{ax} d} \tag{1}$$

$$C_a = 12.2\, A_{663.2} - 2.79\, A_{646.8} \tag{2.1}$$

$$C_b = 21.50\, A_{646.8} - 25.10\, A_{663.2} \tag{2.2}$$

$$C_{x+c} = \frac{(1000\, A_{470} - 1.82\, C_a - 85.02\, C_b)}{198} \tag{2.3}$$

$$C_{ax} = 0.8\, C_{x+c} \tag{2.4}$$

## Shelf life experiments

Different samples were prepared to determine the shelf life of astaxanthin in various matrices and environmental conditions. (1) Free all-$E$-astaxanthin at a concentration of 1 μg/mL in acetone was stored in amber vials at -80°C, -20°C, 4°C, and room temperature for 833 days. Optical density was measured initially, after 11, 42, and 833 days by UV/Vis photometry at λ = 474 nm. (2) Astaxanthin concentration of a concentrated *H. pluvialis* culture (40 g/L) (Sea & Sun Technology) was determined directly and after storage at 4°C in the dark for 104 and 489 days. (3) The same sample was freeze-dried. Therefore, aliquots were poured into small glass vessels with a filling height of approximately 0.5 cm. They were placed into a freeze drier (Alpha 1–4, Christ, Osterode, Germany) for 24 hours at ≤ 37 Pa. After the lyophilization was finished, the vessels were sealed under vacuum and stored at 4°C in the dark. However, one sample was measured directly using the described enzymolytic UHPLC-PDA standard method. The other samples were exposed to ambient air after 7, 108, and 489 days and measured. Afterward, the samples were closed without vacuum sealing and stored under the same conditions until 489 days after lyophilization. Hereafter, all samples were measured again. (4) Two sealed packages of *H. pluvialis* biomass (Golden Peanut) were stored as purchased at -21°C in the dark over the whole experiment. One was opened and closed again tightly at the beginning of the experiment. Both packages were opened after two and a half years and measured using the UHPLC-PDA standard method.

## Statistics

Linear regression was performed by ordinary least squares method, and the significance of the deviation of the y-intercepts from zero was evaluated by t-tests. To determine the device-related measurement deviation, free all-$E$-astaxanthin standards dissolved in acetone were measured regularly prior to or after sample measurements. A total of 45 samples with astaxanthin concentrations between 0 and 45.0 μg/mL were taken into account. Tests on significant deviations were calculated using mean difference tests with a level of significance of σ = 0.05.

## Results and discussion

An enzymolysis-based process for astaxanthin deesterification from natural samples was established to quantify astaxanthin easily without having to identify its various esters. Interactions of sample amount and enzyme quantity were evaluated to find optimal conditions for maximum astaxanthin recovery. These were verified with different astaxanthin-containing samples. Moreover, the shelf life was evaluated to minimize losses prior to analysis. Finally, the method was compared to simple photometric approaches for astaxanthin determination to assess their applicability in process monitoring (Fig 1).

## Calibration curve

**Detection limits and linearity of astaxanthin standards.** For astaxanthin quantification, a calibration was established with free all-*E*-astaxanthin standard dissolved in acetone, filtered, and injected directly into the UHPLC-PDA-MS system in various concentrations. The resulting peak areas were integrated for calibration curve determination. The retention time of the all-*E*-astaxanthin peak was 7.50 minutes. Linear regression resulted in the correlation f(x) = 1693.4x − 990.57 with a coefficient of determination of 0.9985. The coefficient of variation was below 4% for all data points, except for the zero value. The calculated y-intercept was significantly different from zero (Table 1). Measurement of all-*E*-astaxanthin dissolved in acetone has a greater number of replicates than the following measurements because it was also used as a parallel check on the stability of the instrument.

As the process should be applied for natural samples that have to be deesterified, the calibration was repeated with free all-*E*-astaxanthin and astaxanthin monopalmitate. Standards of both substances were subjected to enzymolysis conditions to verify the behavior and linearity of the calibration (Fig 1, a1). The retention time of all-*E*-astaxanthin was 6.43 minutes.

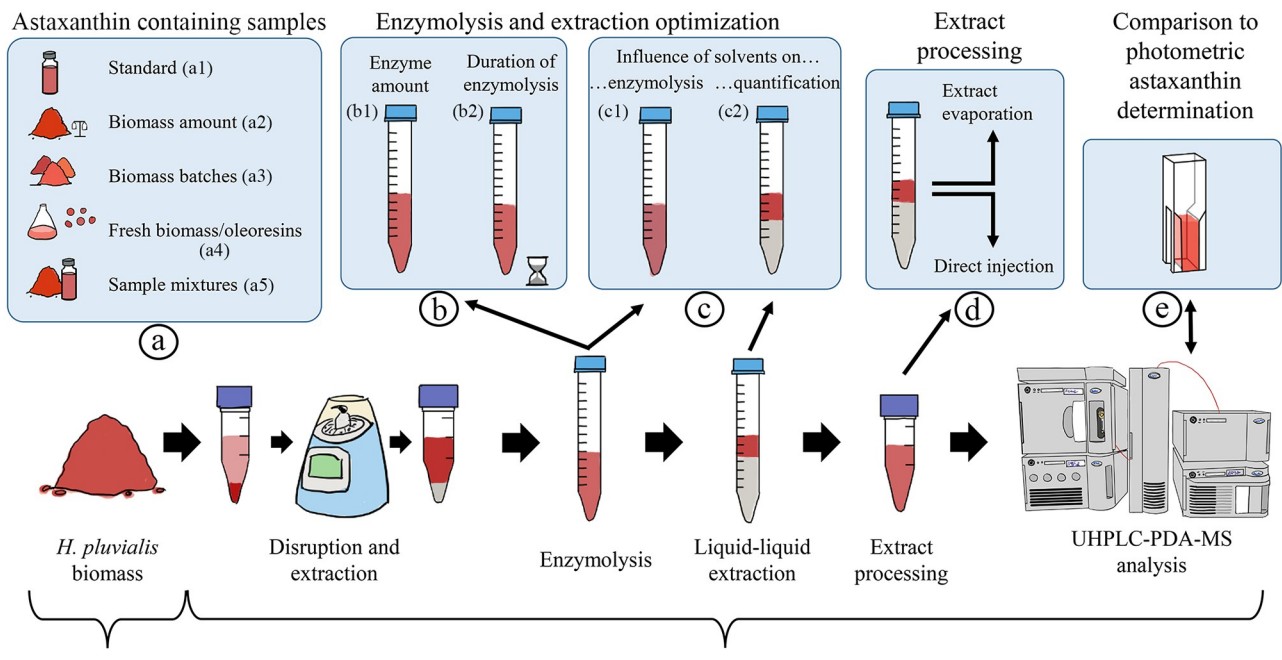

**Fig 1. Schematic overview of the method development for astaxanthin quantification.**

**Table 1. Calibration curves of astaxanthin standards.**

| Sample | | | | Concen-tration limits (μg/mL) | Number of measure-ments | Linear regression | | Linear regression through zero | |
|---|---|---|---|---|---|---|---|---|---|
| Standard | Solvent | Processing | Integrated isomers | | | Equation | R$^2$ | Equation | R$^2$ |
| All-*E*-Ax[a] | acetone | direct | all-*E*-Ax | 0–32.3 | 112 | 1693.4x-990.57* | 0.9985 | 1643.0x | 0.9966 |
| All-*E*-Ax | acetone | direct | all-*E*, 9*Z*, 13*Z*, diZ-Ax | 0–32.3 | 27 | 1718.2x-676.84* | 0.9993 | 1724.4x | 0.9982 |
| All-*E*-Ax | acetone | direct | all-*E*-Ax | 0–11.2 | 78 | 1580.9x-548.95 | 0.9964 | 1493.7x | 0.9917 |
| All-*E*-Ax | acetone | direct | all-*E*, 9*Z*, 13*Z*, diZ-Ax | 0–11.2 | 21 | 1626.2x-429.73 | 0.9962 | **1566.1x** | 0.9933 |
| All-*E*-Ax | PE[c] + acetone | like in enzymolysis | all-*E*-Ax | 0–11.2 | 10 | 1624.4x-65.64 | 0.9996 | 1615.3x | 0.9995 |
| Ax-Mp[b] | PE + acetone | like in enzymolysis | all-*E*-Ax | 0–7.5 | 9 | 1700.8x+5.50 | 0.9994 | 1702.0x | 0.9994 |
| All-*E*-Ax & Ax-Mp | PE + acetone | like in enzymolysis | all-*E*-Ax | 0–11.2 | 18 | 1633.6x+33.13 | 0.9982 | 1638.9x | 0.9982 |
| All-*E*-Ax & Ax-Mp | PE + acetone | like in enzymolysis | all-*E*, 9*Z*, 13*Z*, diZ-Ax | 0–11.2 | 18 | 1753.3x+40.56 | 0.9985 | **1759.9x** | 0.9985 |

Regressions used for quantification are highlighted bold.

*y-intercept significantly different from zero.

[a]Ax = Astaxanthin.

[b]Ax-Mp = Astaxanthin monopalmitate.

[c]PE = Petroleum ether.

Subsequently, calibration was repeated for the obtained astaxanthin UHPLC-PDA peaks and compared to the previous based on the standard dissolved in acetone. Calculations with the previously examined line of best fit did not properly depict the lower concentrations of enzymolyzed astaxanthin in a concentration range between 0.0 to 13.6 μg/mL and resulted in an overestimation of all-*E*-astaxanthin content of a maximum of 131% at a concentration of 0.4 μg/mL and 57% at 1.0 μg/mL astaxanthin. Thus, linear regressions were calculated from the newly obtained values. Regressions resulted in similar linear correlations below 11.2 μg/mL astaxanthin in the liquid-liquid extraction phase, which was the upper detection limit. It was f(x) = 1624.4x − 65.639 (R$^2$ = 0.9996, n = 10) for free all-*E*-astaxanthin from free all-*E*-astaxanthin standard and f(x) = 1700.8x + 5.500 (R$^2$ = 0.9994, n = 9) for free all-*E*-astaxanthin from astaxanthin monopalmitate (Fig 2). The y-intercept was not significantly different from zero for both. As the aim was to quantify astaxanthin derived from natural samples that have to be prepared by enzymolysis, quantification should be performed by using an appropriate calibration curve. The calibration using astaxanthin and astaxanthin monopalmitate subjected to the standard deesterification procedure was the most trustworthy. For higher accuracy, a calibration curve combining the previous two, including the later described isomers and forced through zero, was used. It was f(x) = 1759.9x (R$^2$ = 0.9985, n = 18). This curve represented astaxanthin concentrations between 1.0 μg/mL and 11.2 μg/mL with less than 5% error in both directions.

At 0.5 μg/mL the calculated recovery decreased by 11%. Thus, a concentration between 0.5 and 1.0 μg/mL should be recognized as the lower detection limit. Moreover, the linear correlation ended when the astaxanthin concentration exceeded 11.2 μg/mL. Higher concentrations resulted in reduced recoveries down to 55% at 27 μg/mL. This behavior was correlated to the

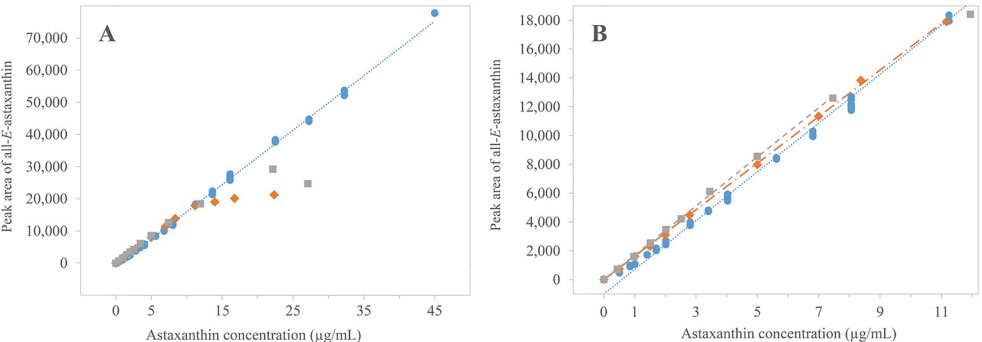

**Fig 2. Calibration curves of differently processed astaxanthin standards.** B close-up from A including the linear regressions through zero of the standards in petroleum ether and acetone mixture. ● all-*E*-astaxanthin standard in acetone ⋯ linear f(x) = 1693.4x-990.57, $R^2$ = 0.9985, ◆ all-*E*-astaxanthin standard in petroleum ether-acetone mixture – – linear f(x) = 1615.3x, $R^2$ = 0.9995, ■ deesterified astaxanthin monopalmitate in petroleum ether-acetone mixture - - - linear f(x) = 1702.0x, $R^2$ = 0.9994.

precipitation of astaxanthin, which was observed at the phase boundary and glass wall when more than 11.2 μg/mL of astaxanthin were present during the liquid-liquid extraction step. The liquid-liquid extraction of astaxanthin into petroleum ether is a delicate step as pure astaxanthin is poorly soluble in pure petroleum ether. During liquid-liquid extraction, acetone is absorbed into the petroleum ether, shifting the equilibrium and accumulating astaxanthin in the upper phase. The solubility of astaxanthin in the petroleum ether-acetone mixture is still limited at an upper boundary of approximately 11.2 μg/mL. This phenomenon was more pronounced for the processed free astaxanthin than processed astaxanthin monopalmitate. E.g., astaxanthin derived from the free standard resulted in a 58% recovery at 22.2 μg/mL, whereas astaxanthin derived from astaxanthin monopalmitate resulted in an 80% recovery at 22.3 μg/mL. This difference indicates an alteration of astaxanthin solubility in the upper layer by the presence of cleaved palmitic acid. Further testing of the type of solvent and ratio might increase astaxanthin solubility, but the functionality of the deesterification process has to be ensured. In the current experimental setup, a concentration of 11.2 μg/mL astaxanthin in the liquid-liquid extraction phase can be considered as the upper detection limit by restricting the amount of biomass for optimal astaxanthin recovery.

The observed shifts in retention time can be attributed to the different solvents. The astaxanthin peak was more than one minute later in pure acetone than in petroleum ether-acetone. This may be due to altered binding behavior on the column at starting conditions and/or miscibility with the mobile phase, which started at a high water content (70% v/v). However, decreasing the water content at the beginning affected peak shape negatively.

**Selectivity.** Considering free all-*E*-astaxanthin standard dissolved in acetone and injected directly, diastereomers were detected besides the main all-*E*-astaxanthin peak (S1 Table). 9*Z*-astaxanthin, 13*Z*-astaxanthin and one di-*Z*-isomer were observed with a medium proportion of total astaxanthin of 2.4±0.2% (n = 29) at 7.90 minutes, 0.4±0.1% (n = 26) at 8.84 minutes, and 0.2±0.03% (n = 26) at 8.56 minutes, respectively. The proportion of 9*Z*- and di-*Z*-astaxanthin remained constant, whereas 13*Z*-astaxanthin rose with prolonged storage and multiple measurements of the same sample up to 2.2%, indicating isomerization reactions during storage in acetone. Organic solvents have been reported to cause the isomerization of carotenoids [54, 66, 67, 102]. These were described to favor the 13*Z*-isomer [67], which can be confirmed here.

In standards processed similarly to enzymolyzed samples, all-*E*-astaxanthin was detected, and two minor peaks were assigned to 9*Z*- (7.51 minutes) and 13*Z*-astaxanthin (8.50 minutes).

The medium proportions of 9Z-astaxanthin, relative to total astaxanthin, were 5.9±0.5% (n = 9) and 4.9±0.3% (n = 8), obtained from free astaxanthin and astaxanthin monopalmitate, respectively. The peak areas of 13Z-astaxanthin were 1.3±0.5% (n = 9) and 2.1±0.9% (n = 8) obtained from free astaxanthin and astaxanthin monopalmitate, respectively. Thus, 9Z- and 13Z-astaxanthin were detected at significantly higher quantities in processed free astaxanthin and astaxanthin monopalmitate than in free astaxanthin dissolved in acetone. Again, this indicates isomerization reactions of the standards during storage and processing. However, 13Z-astaxanthin was not the most abundant diastereomer observed in these experiments, indicating that stereolability is dependent on the specific isomer and conditions [103]. Enzymolysis was performed at 37˚C; elevated temperatures have been reported to enforce isomerization in carotenoids [58, 67, 102, 104–107]. Therefore, the enzymolytic reaction itself bears the potential for further isomerization. These significant differences between the abundance of various isomers under different conditions, especially the solvents used, indicate that a proper calculation of the isomers is complicated. Most astaxanthin determination methods require its solution in one or more solvents and multiple reaction steps, which might shift the proportion of the isomers. Consequently, isomer ratios can be compared within one method to estimate variabilities, but statements beyond cannot be made without additional tests. Changes in the proportions of stereoisomers might be interesting for various applications as diastereomers have been described to exhibit different antioxidant activity *in vitro* [108] and variable bioavailability [79]. Therefore, their exact determination should be studied further.

For astaxanthin quantification, in the following experiments, the linear regression of the combined results of the enzymolyzed free-astaxanthin and astaxanthin monopalmitate was used as described above. To account also for the isomers, their peak areas were included. Therefore, their peak areas were multiplied by correction factors to integrate their different extinction properties. These were 1.20 for 9Z-astaxanthin, 1.56 for 13Z-astaxanthin, and 1.11 for the di-Z-astaxanthin isomers based on the extinction coefficients of Bjerkeng et al. [79]. All obtained areas were summed and the resulting calibration curve was f(x) = 1759.9x ($R^2$ = 0.9983, n = 18). All different diastereomers were also found in extracts from *H. pluvialis* (Fig 3).

It can be concluded that the amount of astaxanthin is the limiting factor for enzymolysis or liquid-liquid extraction. A sensible measurement range is 1.0 to 11.2 μg/mL of total astaxanthin concentration in the extraction phase. Moreover, applying the respective calibration curve is essential to keep the quantification error below 5%.

## Method development

Method development was based on dried *H. pluvialis* powder. All relevant process steps, i.e., disruption and extraction, enzymolysis, liquid-liquid extraction, and the processing of the liquid-liquid extracts, were investigated and adapted if needed. Subsequently, detection limits, linearity, and precision of the obtained method were determined (Fig 1).

**Disruption of *H. pluvialis* and astaxanthin extraction.** Astaxanthin was extracted from *H. pluvialis* biomass by disrupting the cells with a bead mill in the presence of acetone. This process was superior to grinding and ultrasonication under various conditions. A complete decolorization of the cell pellet, as an indicator for complete astaxanthin extraction, was only achieved by bead milling.

**Enzymolysis.** The enzymolysis is the core of the established method. Proper enzyme function and turnover must be ensured for complete deesterification of the various astaxanthin esters and free astaxanthin recovery. To find an optimal enzyme concentration, its amount and incubation time were varied, and free astaxanthin was measured (Fig 1b).

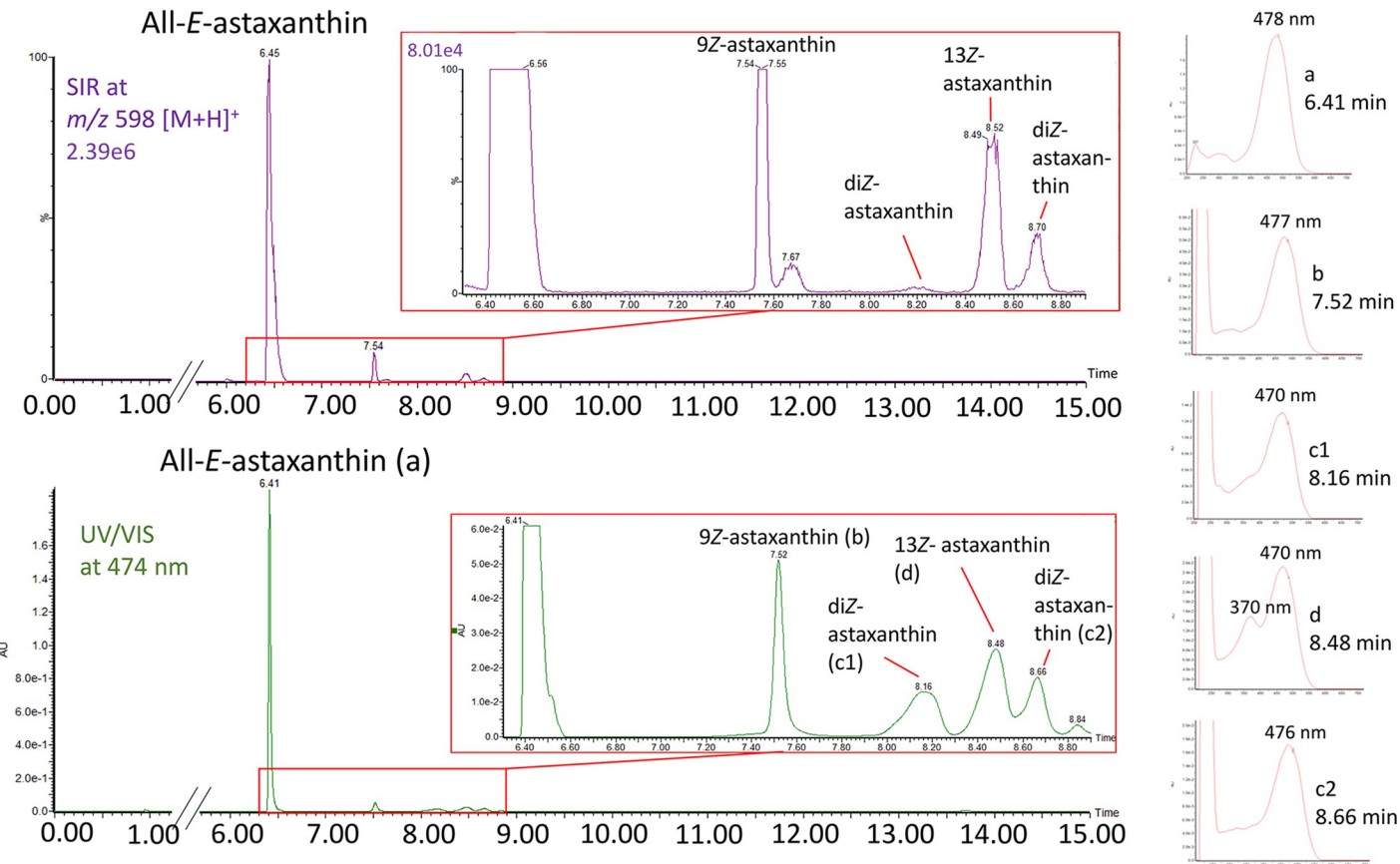

**Fig 3. UV/Vis and SIR chromatograms of an enzymolyzed *H. pluvialis* extract and corresponding absorption spectra of astaxanthin diastereomers.**

The variation of cholesterol esterase (Fig 1, b1) from 0.05 to 2.00 units added to a constant quantity of 0.4 mg *H. pluvialis* biomass per sample resulted in calculated total astaxanthin proportions between 0.87±0.30% w/w (n = 3) and 4.20±0.03% w/w (n = 3) (Fig 4 and S1 Table). Respectively, 0.003 units to 0.12 units had been present per µg total free astaxanthin. The resulting all-*E*-astaxanthin concentrations in the petroleum ether-acetone phase were between 1.33 and 6.29 µg/mL, thus in the linear calibration range. The increasing astaxanthin content with a higher enzyme concentration indicates substrate conversion by cholesterol esterase in the presence of acetone. Cholesterol esterase has been shown to work in a wide pH- and temperature range and to maintain its activity in the presence of solvents [109]. Furthermore, an activity increase was reported when minor proportions of up to 10% v/v of organic solvents were added [110]. Thus, the enzyme seems appropriate for astaxanthin deesterification in the described solvent-rich environment. The lowest astaxanthin values, accompanied by a distinct increase in the calculated astaxanthin concentration and comparatively high standard deviations (0.13–0.57% w/w), were observed between 0.05 and 0.50 units of cholesterol esterase. Here, enzymolysis was incomplete. Astaxanthin content recovered from treatment with 1.0 unit cholesterol esterase did not differ significantly from 1.5 units. Still, 2.0 units resulted in significantly more total astaxanthin, however, with a small effect size compared to 1.5 units (0.16 percentage points). Therefore, 1.0 to 2.0 units hydrolyzed the majority of astaxanthin esters. Su et al. reported good deesterification results with 4.0 units of cholesterol esterase per reaction. However, they did not specify the used biomass amount [85]. For lipases, slower

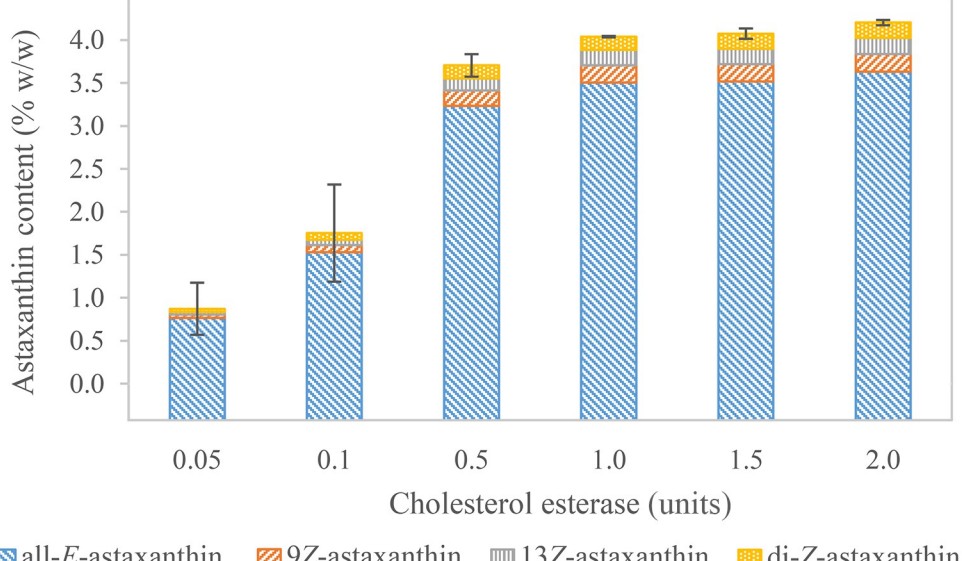

**Fig 4. Astaxanthin content in 0.4 mg *H. pluvialis* biomass, enzymolyzed with varying amounts of cholesterol esterase.** Standard deviation is indicated for total astaxanthin in triplicates; 1.0 unit was a duplicate. Standard deviations for the individual isomers can be found in S1 Table.

conversion rates and smaller efficiencies have been reported. 4.6 U/µg carotenoids have resulted in 63.2% free astaxanthin recovery after 7 hours of incubation. Moreover, the recovery even decreased when more enzyme was applied [93]. Huang et al. showed the highest free astaxanthin yields of 80% with 80 units per µg of astaxanthin esters with a recombinant lipase after one hour of incubation [94]. Working with higher enzyme amounts may secure complete conversion, especially at higher astaxanthin concentrations. Nevertheless, astaxanthin solubility in subsequent liquid-liquid extraction is limited, and higher enzyme application is not necessarily more beneficial. Consequently, 0.06 to 0.12 units per µg of total free astaxanthin were considered sufficient for most samples. This corresponded to 1.0 to 2.0 units of cholesterol esterase per sample or 2.5 to 5.0 units per mg of *H. pluvialis* biomass in these experiments. A further optimization might be achieved by using a higher enzyme concentration and simultaneously repeating the liquid-liquid extraction process as performed by Moretti et al. [61].

9*Z*-, 13*Z*- and two di-*Z*-isomers of astaxanthin were observed in all samples. Their biomass proportions were 0.21±0.01% w/w (n = 3), 0.19±0.02% w/w (n = 3), and 0.18±0.01% w/w (n = 3), respectively, at 2.0 units. Regarding their proportions to total astaxanthin, no significant difference was observed in the di-*Z*-isomers when cholesterol esterase concentration was varied. Increasing cholesterol esterase from 0.05 to 2.0 units caused the proportions of 9*Z*- and 13*Z*- to total astaxanthin to rise from 4.35 to 4.89% and from 3.56 to 4.47%, respectively. 9*Z*-astaxanthin was equally or less abundant than in the enzymolyzed standards, whereas 13*Z*-astaxanthin and the di-*Z*-isomers were detected in higher proportions. Higher or equal diastereomer concentrations should be assumed in natural samples than in pure all-*E*-astaxanthin standards. The lack of 9*Z*-astaxanthin in this natural sample might indicate that sample composition influences the isomer equilibria, as it is more complex than the standards. Further isomerization to other isomers or even reverse isomerization to all-*E*-astaxanthin [103, 111] during enzymolysis and extraction is possible. Compared to the standards, a higher proportion of 13*Z*-astaxanthin and its di-*Z*-isomers might be assumed in this sample. However, their abundance might partially also arise from isomerization reactions. Besides the already

mentioned effects of temperature and solvents on isomerization, the sample matrix is more complex due to other cell components from *H. pluvialis*. NaCl and iodine have also been described to catalyze the isomerization behavior of carotenoids [67, 79, 103]. Therefore, reliable quantification of their total amounts is not possible. These results can only be seen as an estimate for the calculation of total astaxanthin.

The duration of enzymolysis was extended from 0.75 to 1.5 hours. 0.5 and 2.0 units of cholesterol esterase were applied to constant biomass of 0.4 mg dried *H. pluvialis* (Fig 1, b2). No significant difference in the measured all-*E*-astaxanthin amount was observed compared to the shorter incubation time at both enzyme concentrations. This indicates that increasing enzyme concentration is more efficient than elongating incubation time. Additionally, studies showed a tendency of free astaxanthin to decrease when enzymolysis was prolonged [85], which further encourages shorter incubation at higher enzyme concentrations. With longer incubation time, the proportion of 9*Z*-astaxanthin and the di-*Z*-isomers decreased by 9–18%, whereas 13*Z*-astaxanthin remained constant. Degradation and/or isomerization of astaxanthin isomers during and after deesterification is possible but seems less severe compared to alkaline saponification [85]. Although longer enzymolysis duration may still result in good all-*E*-astaxanthin recoveries, as also shown by Su et al. [85], relatively higher cholesterol esterase levels at shorter incubation time can result in sufficient total astaxanthin yields without the risk of further isomerization reactions. 2.0 units of cholesterol esterase and 0.75 hours of incubation time can be used as an initial approach toward the astaxanthin measurement of an unknown sample, which can be adapted if necessary. Further improvement of enzymolysis might be achieved by surface-active detergents or bile acids, as indicated by Uwajima et al. [109], but might be limited due to interaction with the used solvents.

**Liquid-liquid extraction.**   After enzymolysis, free astaxanthin had to be recovered from the buffer solution. Therefore, liquid-liquid extraction with 2 mL of petroleum ether was performed. After phase separation, the upper layer increased in volume (Fig 1, c2). This volume was determined at 2.31+0.03 mL (n = 4). Acetone was dissolved in petroleum ether during phase mixing, resulting in an excess volume [112]. Various studies encountered similar problems by drying and resolving the combined extracts in solvent [61, 83]. This procedure is more laborious and prone to astaxanthin losses. As the volume of the petroleum ether phase is crucial for the exact quantification of astaxanthin, the increased volume was used to calculate astaxanthin content in all samples processed likewise. Moreover, sodium sulfate has been used [61, 83] to remove residual water from the petroleum ether phase and enhance solvent strength. However, this is unnecessary when the described boundaries for astaxanthin solubility are observed and dry or highly concentrated samples are used.

**Processing of liquid-liquid extracts.**   The extract obtained from liquid-liquid extraction can be injected directly into UHPLC-PDA-MS or further processed (Fig 1d). Various authors evaporated the upper extraction phase and redissolved the extracts in organic solvents prior to measurement [61, 83]. These methods were compared to evaluate recoveries and find the most cost and time extensive procedure. Direct analysis of this phase resulted in a recovery of 98.24 ±1.43% (n = 4) total astaxanthin. Further processing resulted in a decrease in recovery. An aliquot was evaporated and redissolved in the same volume of acetone. Its astaxanthin content was quantified with the respective calibration for astaxanthin in acetone. Total astaxanthin recovery was significantly lower with 91.68±1.32% (n = 4). A repeated extraction, evaporation, and resumption in acetone resulted in 96.34±0.68% (n = 3) total astaxanthin recovery. The generally lower recovery might be due to losses during processing. The proportion of 9*Z*-astaxanthin was lower, whereas 13*Z*-astaxanthin and the di-*Z*-isomers were more abundant when the samples were redissolved in acetone (S1 Table). This direct comparison and comparison to the standards for calibration curve determination demonstrate changing isomer recoveries in

different solvents, indicating once more that a valid determination is not possible due to changes during processing. Altogether, these experiments point to direct processing as the most correct and time and cost-reducing procedure. Still, the quantification of the diastereomers in natural samples of *H. pluvialis*, especially in the presence of other carotenoids, is more precise when the samples are dissolved in acetone, as the chromatographic resolution of the diastereomers and lutein was higher.

**Detection limits and linearity of astaxanthin determination.** The complex composition of *H. pluvialis* biomass was assumed to influence enzymolysis. Moreover, the previously determined detection limits should be verified when applying natural samples. Therefore, the biomass quantity of *H. pluvialis* was varied from 0.04 to 3.98 mg at constant saponification time (0.75 hours) and cholesterol esterase concentration (2.0 units) [99] to find an optimal astaxanthin transformation range (Fig 1, a2). A maximum yield of 3.74±0.01% w/w (n = 3) all-*E*-astaxanthin was observed at 0.8 mg of *H. pluvialis* powder (≙ 13.0 μg/mL all-*E*-astaxanthin concentration in liquid-liquid extract) (Fig 5 and S1 Table). Between 0.2 and 2.0 mg (≙ 3.18 and 32.4 μg/mL all-*E*-astaxanthin concentration in liquid-liquid extract), ≥ 97% of the maximum recovery were still reached. This implies an extended measurement range compared to the standards, whose recovery decreased significantly, starting at approximately 11.2 μg/mL astaxanthin concentration. The sample matrix of whole-cell biomass is much more complex than the standards. Lipophilic cellular components such as fatty acids or other carotenoids are dissolved in the hydrophobic solvent, altering its composition and enhancing solubility for other constituents like astaxanthin. *Z*-isomers of many carotenoids, including astaxanthin, have a higher solubility in various solvents than the all-*E*-isomer [104, 113, 114]. Thus, the calibration curve might be applied at astaxanthin concentrations above 11.2 μg/mL. However, a definite range cannot be specified because biomass composition, especially in terms of fatty acids and pigments, is variable, particularly when *H. pluvialis* is exposed to stress conditions [88, 115–117]. The measured all-*E*-astaxanthin content decreased significantly, applying either more or less biomass. It was 3.33±0.05 (n = 3), 3.24, and 2.86% w/w for 0.04, 3.26, and 3.98 mg of biomass, respectively (≙ 0.64, 51.81, and 63.25 μg/mL all-*E*-astaxanthin concentration in

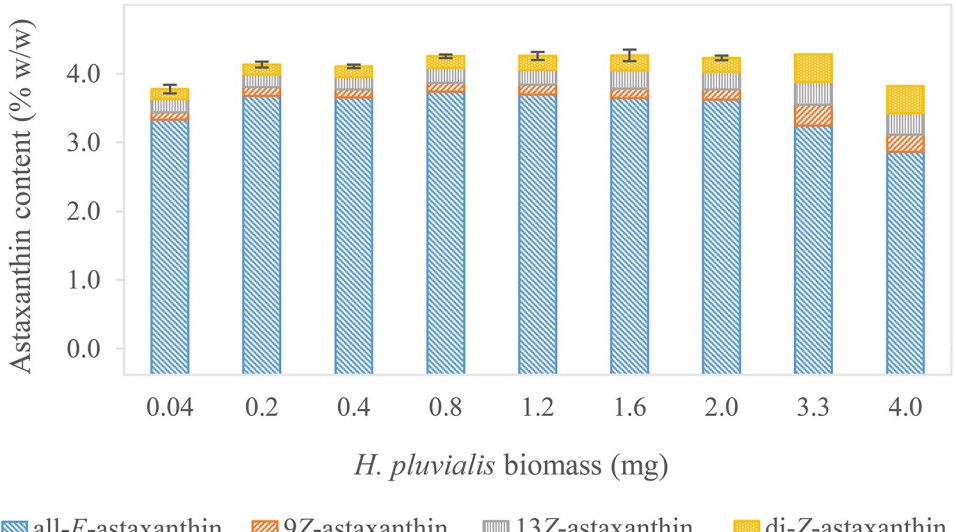

**Fig 5. Astaxanthin content in *H. pluvialis* biomass, deesterified with 2.0 units of cholesterol esterase.** Standard deviation is indicated for total astaxanthin in triplicates. Standard deviations for the individual isomers can be found in S1 Table. The figure demonstrates that the method gets similar results for a broad range of biomass inputs.

liquid-liquid extract). The decrease at 0.04 mg is likely due to measuring inaccuracy at the lower border of the measurement range, which was also determined in the standards. The significant decrease in recovery when using more than 2.0 mg of biomass might be explained by a relatively too low enzyme concentration. Moreover, other carotenoid and cholesterol esters and phospho- and triacylglycerides are in direct competition with the deesterification of astaxanthin-esters and may decelerate the process [118–121]. Jacobs et al. reported higher conversion for carotenoid esters that contain a cyclopentenyl terminal ring rather than a cyclohexenyl terminal ring, implying other carotenoid esters might be hydrolyzed preferably [91]. Moreover, on the fatty acid side, a higher hydrolysis rate has been shown for longer chain and polyunsaturated fatty acids esterified with cholesterol [109]. This might further favor reactions of the enzyme with other molecules. Additionally, proteins might inhibit proper enzyme function [122–124]. Another reason might be insufficient solubility of astaxanthin in the extraction phase. However, for none of the samples, precipitation was visible. The proportion of 9$Z$- and 13$Z$-astaxanthin increased significantly only when relating the samples with the lowest and highest biomass (0.04 and 2.0 mg). However, it did not vary significantly when samples with similar concentrations were compared. Moreover, the proportion of all diastereomers increased when the biomass amount was raised to 3.3 and 4.0 mg. These corresponded to all-$E$-astaxanthin concentrations of 51.81 and 63.25 µg/mL, which are well above the upper detection limit. Thus, the higher solubility of the $Z$-isomers [104, 113, 114] might change the equilibrium in their favor.

This experiment expanded the outlined upper detection limit of 11.2 µg/mL of all-$E$-astaxanthin in the extraction phase to approximately 30 µg/mL. The lower detection limit was approved at approximately 1.0 µg/mL.

**Precision of astaxanthin determination.** In order to demonstrate measuring precision and repeatability, astaxanthin content was measured multiple times in two different samples. Each measurement was performed on a different day to prove the similarity of the independent experiments. The samples had an all-$E$-astaxanthin content of 3.46±0.04% w/w (n = 5) and 3.98±0.04% w/w (n = 4) (Fig 6 and S1 Table). This equaled coefficients of variation of

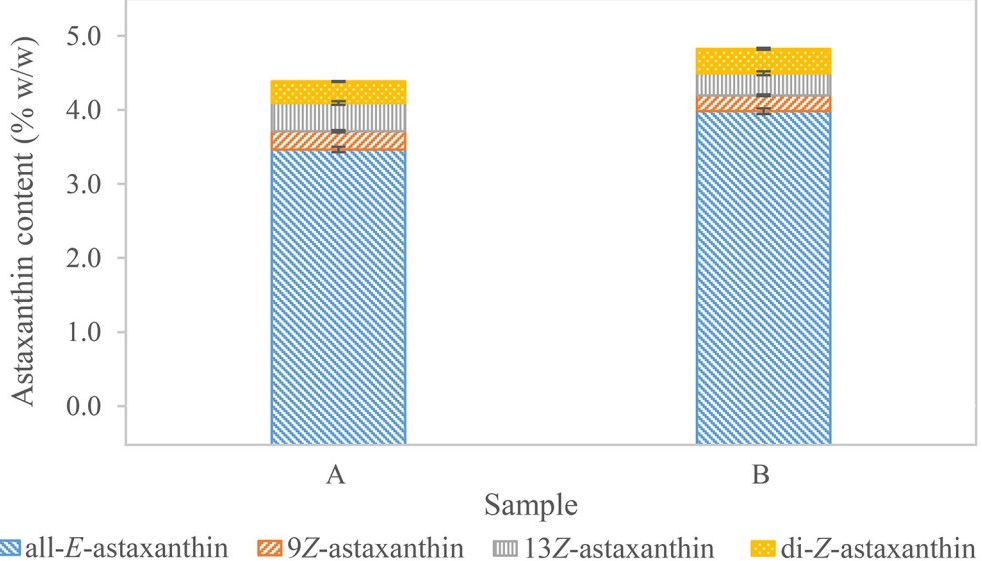

**Fig 6. Astaxanthin content in two *H. pluvialis* powders.** Standard deviation is indicated for each astaxanthin diastereomer. Sample A: n = 5, sample B: n = 4.

1.1% and 1.0%, respectively. The isomers 9Z-, 13Z-astaxanthin, and two di-Z-isomers were observed in both samples with coefficients of variation between 1.9 and 9.0%. Thus, the precision of all-E-astaxanthin measurement was high, whereas it fluctuated for the diastereomers. It might be improved by evaporating and resolving the extracts in acetone prior to analysis. However, such treatment would be most likely at the expense of total astaxanthin recovery.

## Method adjustment

The resulting method was further validated regarding linearity, detection limits, precision, and robustness when various samples of astaxanthin containing biomass or extracts were used. Therefore, fresh *H. pluvialis* cultures and either pure oleoresins or oleoresins diluted in ethanol were applied at various concentrations. The sample type specific method adjustments and validations are explained in the following.

**Sample type: *H. pluvialis* liquid cultures.** *H. pluvialis* cells concentrated in nutrient-depleted medium were disrupted in the presence of acetone. The first passage in the bead mill had only little effect on the extraction of carotenoids, as hardly any color change of the medium-acetone phase was observed. Major cell disruption and extraction were achieved only in the second and third passage. This might be due to an increased acetone proportion during the passages, which has a stronger dewatering effect on membranes and cell walls.

Biomass of four batches of *H. pluvialis* was processed according to the standard method (Fig 1, a4). All-E-astaxanthin content was determined ranging from 0.73±0.04% w/w (n = 3) to 3.82±0.15% w/w (n = 3) (Fig 7 and S1 Table). Independently from the actual astaxanthin concentration, the obtained all-E-astaxanthin recovery in three of the four batches was above 90% of the maximum achieved concentration when using 0.2 to 1.5 mg of biomass, respectively 2.5 to 15.0 µg/mL of all-E-astaxanthin extract concentration. These results confirm the applicability of the linear calibration correlation of all-E-astaxanthin at concentrations above 11.2 µg/mL for natural samples. However, the maximum all-E-astaxanthin recovery was observed at different extract concentrations in each sample. These extended over the whole measurement

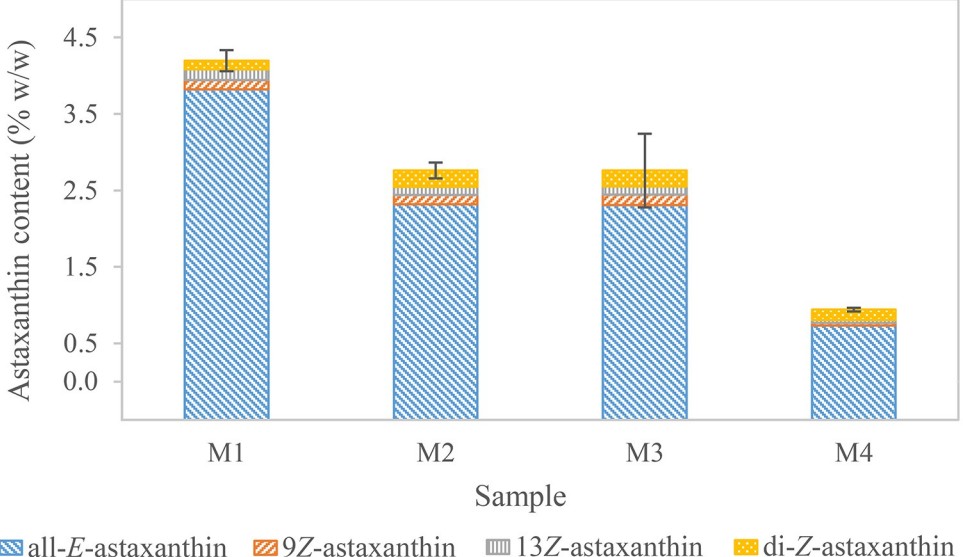

**Fig 7. Astaxanthin content of four batches of *H. pluvialis* suspended in residual medium.** M1 to M4 derive from different batches. Standard deviations are indicated for total astaxanthin. Replicates of M1 to M4 vary between three and five. Standard deviations for the individual isomers and replicates can be found in S1 Table.

range and were found at 2.5, 8.2, 11.3, and 19.7 μg/mL. Thus, the previously determined measurement range should be condensed to an optimal measurement range, which is highly dependent on the sample. Throughout its life cycle, *H. pluvialis* exhibits different characteristics, e.g., highly stressed cells with a high astaxanthin level also indicate an altered composition regarding fatty acids, carotenoids, and other cell compounds [44, 68, 88, 115–117, 125], which might further change enzyme activity, solvent equilibria, and astaxanthin solubility.

In most samples, the measured isomers essentially showed the same proportion when between 0.75 and 1.75 mg of *H. pluvialis* biomass (dry weight) were used. Applying less than 0.75 mg biomass resulted in higher deviations of the measurements. This reinforces the assumption that the proportion of isomers measured is largely stable within a constant experimental setup. In relation to the total astaxanthin content, 2.65 to 14.54% di-*Z*-astaxanthin, 2.96 to 5.03% 9*Z*-astaxanthin, and 3.38 to 4.17% 13*Z*-astaxanthin were observed. Yuan and Chen, who compared the different isomers in extracts of *H. pluvialis* during and after saponification under alkaline conditions, observed a higher peak area ratio of 9*Z*- to 13*Z*-astaxanthin [55–57]. Gong et al. also observed higher 9*Z*- than 13*Z*-astaxanthin proportions after enzymolysis [125], indicating a higher abundance of 9*Z*-astaxanthin in general. However, changes in the composition of isomers have been observed during *H. pluvialis* cultivation [125], leading to the suggestion that strain, cultivation, and stress conditions influence the abundance of the different isomers, as evidenced by the high variance of di-*Z*-isomers in these experiments. As already discussed, storage and processing conditions might also have an influence, and the comparison to other methods is presumably biased.

Compared to the share of geometrical isomers observed in the deesterified all-*E*-astaxanthin standards, 9*Z*-astaxanthin was less or equally abundant, and 13*Z*- as well as the di-*Z*-astaxanthin isomers were more abundant in samples of *H. pluvialis* suspended in medium. Thus, the same principle that biomass, temperature, and solvents affect isomerization behavior applies as already described previously for dried *H. pluv*ialis samples.

**Sample type: Oleoresins.** Oleoresins from *H. pluvialis* are mostly obtained by SC-CO$_2$ extraction. Here, modifiers are often used for enhanced extractability [89, 126–128] and due to system requirements. Ethanol, as a solvent, is very common and can be found in the resulting extracts. For astaxanthin determination, it can be evaporated prior to enzymolysis. For cost and time reduction, direct measurement of ethanolic extracts was investigated using the method developed here. Its robustness and effects on further processing and astaxanthin recovery were examined.

The proper function of the enzymolysis in the presence of ethanol had to be ensured (Fig 1, c1). No significant difference was observed between the samples processed with ethanol and those processed without ethanol with respect to all-*E*-astaxanthin. An extract with a high astaxanthin concentration resulted in 52.9±0.4μg/mL (n = 5) of all-*E*-astaxanthin when equal or less than 18% v/v ethanol were present during enzymolysis and extraction. 53.1±0.5 μg/mL (n = 2) of all-*E*-astaxanthin were detected when ethanol was evaporated prior to processing (S2 Table). Another extract with a lower astaxanthin concentration resulted in 8.6 ±0.2 μg/mL (n = 3) of all-*E*-astaxanthin when equal or less than 18% v/v ethanol was present during deesterification. Comparably 8.8±0.2 μg/mL (n = 2) of all-*E*-astaxanthin were obtained when ethanol was evaporated prior to processing. As already described, cholesterol esterase has been shown to maintain its activity in the presence of solvents and, more specifically, ethanol [109]. It might even work better if small amounts of up to 10% v/v of organic solvents are added [110]. Ethanol concentrations above 18% v/v resulted in a sharp decline of the measured all-*E*-astaxanthin content. It decreased to 50% and 16% of the previously calculated astaxanthin quantities for 36% v/v and 54% v/v of ethanol, respectively. A decrease in activity and stability with $\geq$ 10% v/v ethanol concentrations has been observed [110]. A

similar effect has been shown for a lipase from *P. aeruginosa* [95]. It can be explained by stabilization and destabilization of hydrophobic interactions depending on ethanol concentration [129–131], suggesting an inhibition of the enzyme with ethanol concentrations ≥ 18% v/v in these experiments. There was no significant difference in the proportion of 13*Z*-astaxanthin comparing the deesterification in the presence and without ethanol. However, without ethanol, the proportion of 9*Z*-astaxanthin was 18 and 40% higher in the samples with a higher or lower total astaxanthin content, respectively. In the sample with less astaxanthin, the proportion of the di-*Z*-astaxanthin isomers was 60% higher in the absence of ethanol. Solvent-related isomerization might account for this. Overall, the all-*E*-astaxanthin amount did not change significantly. Thus, minor ethanol proportions in the enzymolytic process can be regarded as unproblematic.

After enzymolysis, in the liquid-liquid extraction, the upper phase expansion was studied in the presence of various quantities of ethanol (Fig 1, c2). After processing and extracting with petroleum ether, the upper-phase volume decreased to 2.24, 2.17, 2.11, 2.04, and 2.00 mL when 3, 7, 11, 14, and 18% v/v ethanol were used, respectively. Linear regression resulted in a line of best fit with f(x) = -0.0003x + 2.3052 ($R^2$ = 0.9943, n = 6). Binary mixtures of hexane and ethanol cause a volume dilatation [132], whereas mixtures of acetone and ethanol result in volume contraction [133, 134]. This second phenomenon might also impact the ternary mixture of petroleum ether, acetone, and ethanol, resulting in a smaller excess volume than in the absence of ethanol [135]. When 36% v/v ethanol were initially added to the sample, this linear correlation no longer applied as the volume of the upper phase decreased to 1.95 mL. Here the influence of a higher ethanol proportion resulted in a volume contraction.

It can be concluded from both experiments that the presence of ethanol generally does not influence the total astaxanthin recovery as long as its effect on the volumetric change in liquid-liquid extraction is taken into account during quantification and the volumetric maximum of 18% v/v is considered.

To study the optimal concentration range, detection limits, and linearity of astaxanthin determination in oleoresins (Fig 1, a4), three different samples were deesterified with 2.0 units to 4.0 units of cholesterol esterase at quantities between 0.15 and 15.8 mg. Above 90% of the maximum measured astaxanthin content were recovered when using 0.3 to 1.2 mg of oleoresin. This corresponded to 1.8 μg/mL to 20.4 μg/mL of all-*E*-astaxanthin in the respective extract. There was a sample-specific optimal measuring range. Exceeding or falling below it resulted in reduced recoveries.

In oleoresin O1, the highest all-*E*-astaxanthin value of 5.8% w/w was observed at 0.4 mg oleoresin (≙ 10.0 μg/mL all-*E*-astaxanthin in the liquid-liquid extract) enzymolyzed with 4.0 units of cholesterol esterase. Stepwise increase of oleoresin to 1.6 mg (≙ 41.2 μg/mL all-*E*-astaxanthin) and reducing the enzyme to 2.0 units resulted in a steady 17% decrease of overall all-*E*-astaxanthin recovery. This indicates an excess of enzyme capacity and/or solubility at higher oleoresin amounts. Oleoresins mainly contain fatty acids and other lipophilic compounds such as carotenoids and their esters [39, 101, 136], which can act as competitive substrates.

In oleoresin O2, which was oleoresin O1 diluted in sunflower oil, the highest all-*E*-astaxanthin levels were observed between 0.75 and 1.00 mg of oleoresin (≙ 4.4 to 5.9 μg/mL all-*E*-astaxanthin in the liquid-liquid extract). Less astaxanthin was detected when applying more or less oleoresin. A doubling of enzyme concentration in sample O2 resulted in a minor increase of maximum 5.5% all-*E*-astaxanthin or 3.9% total astaxanthin when directly comparing samples with equal or less than 1.0 mg (≙ 5.9 μg/mL in the liquid-liquid extract) oleoresin. Therefore, the enzyme capacity was sufficient to deesterify astaxanthin in the range below 5.9 μg/mL.

Conversely, in sample O3, the highest all-*E*-astaxanthin value of 3.6% w/w was observed at the highest applied oleoresin per sample, which was 1.65 mg (≙ 25.9 μg/mL all-*E*-astaxanthin

in the liquid-liquid extract) and decreased to 3.0% w/w at 0.78 mg oleoresin ($\triangleq$ 12.3 µg/mL all-$E$-astaxanthin in the liquid-liquid extract). This supports the hypothesis that complex lipophilic samples can be used at concentrations exceeding the previously determined measurement range. Possibly the higher abundance of lipophilic substances results in enhanced solubility of astaxanthin. For proper enzymatic activity, the previously outlined maximum astaxanthin concentration of approximately 30 µg/mL should still be considered, or enzyme quantity has to be increased.

No severe influence of the amount of oleoresin used on the proportion of the isomers was observed in the range below 1.8 mg of sample. 7.51 to 19.69% of the total astaxanthin were the di-$Z$-forms, 5.14 to 13.01% 9$Z$-astaxanthin, and 3.99 to 7.24% 13$Z$-astaxanthin. Their abundance was higher than in the alga cultures suspended in medium or dried cells. Two effects might account for this. First, elevated temperatures and high pressures, also present during SC-$CO_2$ extraction, lead to a higher isomerization rate and altered isomer profiles [58, 102, 104–107]. However, SC-$CO_2$ extraction is a process that is considered mild. Álvarez et al. did not find significant differences in isomer proportions between different extraction conditions [137]. Second, $Z$-isomers might have higher extraction rates in SC-$CO_2$ extraction a priori due to their higher solubility in solvents [104, 113, 114]. This might be the reason for the selective accumulation of $Z$-isomers [138, 139].

In this study, significant higher proportions of 9$Z$- than 13$Z$-astaxanthin were detected in all oleoresins (Fig 8 and S1 Table). Various authors have also reported this in enzyme deesterified supercritical fluid extracts of *H. pluvialis* [85, 137]. All-$E$-astaxanthin and all-$E$-astaxanthin diacetate exposed to isomerization inducing conditions showed higher levels of 9$Z$- and 13$Z$-astaxanthin than other di-$Z$-isomers in the resulting mixtures [58, 60].

**Sample mixtures.** To further quantify the observed effects of natural samples on the solubility and thus linear measurement range of astaxanthin in liquid-liquid extracts, different standards and a natural sample with equal astaxanthin amount were examined independently and in mixtures (Fig 1, a5). The all-$E$-astaxanthin recoveries of the independent samples were

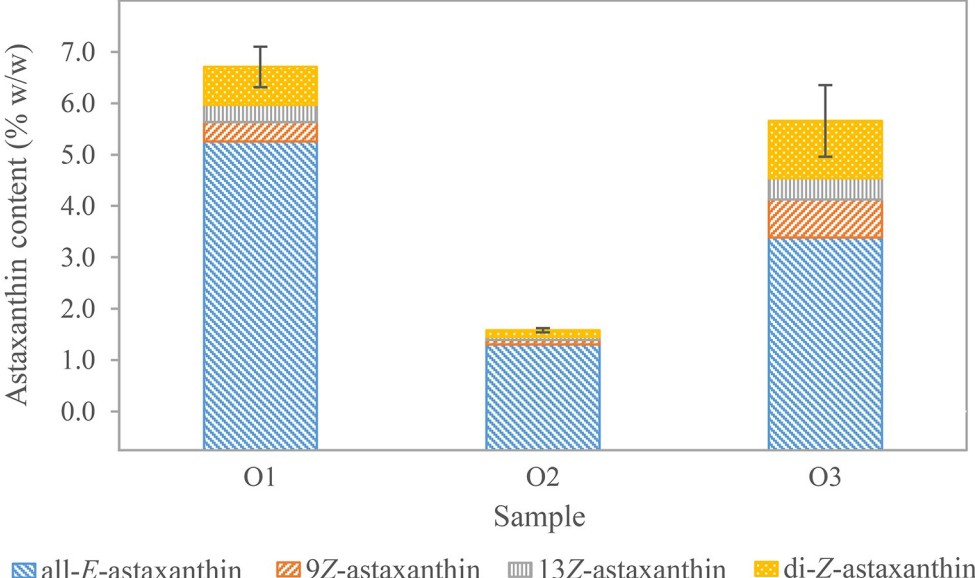

**Fig 8. Astaxanthin content of three different oleoresins of *H. pluvialis*.** Standard deviations are indicated for total astaxanthin. O1 (n = 3) and O3 (n = 3) derive from different batches. O2 (n = 7) is a dilution of O1. Standard deviations for the individual isomers can be found in S1 Table.

88% from astaxanthin monopalmitate, 83% from free astaxanthin, and 98% from *H. pluvialis*. The calculated astaxanthin concentration in the final extraction phase was between 4 and 5 μg/mL. The addition of acetone after liquid-liquid extraction might have caused the unexpected reduced recovery of the standards. Mixtures of those samples yielded recoveries of 95% to 109%, even at a total all-*E*-astaxanthin concentration of 13.3 μg/mL, indicating that the presence of biomass components or acetone facilitates the solution of astaxanthin in the extraction phase. Evaluating all these samples, a proportional increase of the measured astaxanthin concentration to the prepared astaxanthin concentration was observed: a linear regression forced through zero resulted in a proportion of 1.0205 of measured to prepared total astaxanthin concentration with a coefficient of determination of 0.9654. However, the recovery decreased when free all-*E*-astaxanthin standard dissolved in acetone was added after the enzymolysis. The same samples prepared with 3.6 μg and 7.2 μg free all-*E*-astaxanthin standard resulted in a correlation of 0.9408 ($R^2 = 0.9690$) and 0.8836 ($R^2 = 0.9441$), respectively. The addition of astaxanthin and acetone after extraction might change the phase equilibria, density, and astaxanthin solubility, resulting in the observed measurement inaccuracy. Apart from these relations, all-*E*-astaxanthin extract concentrations up to 23.6 μg/mL were detected linearly without precipitation effects. Moreover, those samples containing the whole-cell biomass or mixtures resulted in higher recoveries in all experiments. Thus, fatty acids and other cellular components are the only other differing factors in these experiments that might influence astaxanthin solubility. It is difficult to discriminate between the two observed effects of reduced and enhanced recovery due to acetone/standard addition and the presence of cellular components, respectively, due to the experimental setup. To understand the circumstances affecting all-*E*-astaxanthin solubility, recovery, and thus measurement boundaries for its quantification, a broader concentration range without further astaxanthin addition after processing needs to be considered, as all previous tests showed that the applicable measurement concentration of all-*E*-astaxanthin in *H. pluvialis* biomass is higher than in measurements of standards.

## Method comparison to photometric astaxanthin estimation

Many methods for fast and easy astaxanthin determination are based on a simple photometric measurement of *H. pluvialis* extracts. Two such approaches using different mathematical models were compared to the developed UHPLC-PDA method to evaluate the discrepancy between them and conclude whether photometric techniques can still lead to a reasonable estimate of astaxanthin (Fig 1e). For an even broader impression of the possible validity of those methods, two *H. pluvialis* cultures were tested at different times towards the end of a cultivation period to account for possible changes in the carotenoid and chlorophyll composition. Astaxanthin production had been induced by exposure to high light intensities and nitrogen starvation. Besides astaxanthin, the lutein content was determined by UHPLC-PDA measurements, and spectra of the extracts were compared. UHPLC data revealed steadily increasing all-*E*-astaxanthin content from 0.99 to 2.47% w/w and 1.31 to 2.70% w/w in batch A and B, respectively (Fig 9 and S1 Table). Total astaxanthin concentration rose from 1.13 to 2.79% w/w and from 1.50 to 3.07% w/w in batch A and B, respectively. Photometric data evaluated with the general equation (Eq 1) resulted in total astaxanthin contents between 1.34 and 2.85% w/w and between 1.97 and 2.95% w/w in batch A and B, respectively. Astaxanthin content calculated from the same photometric data but with the equations of Lichtenthaler et al. [100] increased from 1.31 to 2.78% w/w in batch A and from 1.91 to 2.93% w/w in batch B. Astaxanthin amounts obtained from the photometric methods were generally similar. Though, here a higher astaxanthin content was estimated compared to UHPLC-PDA measurements in the first half of both experiments. On the 27th and 28th day, photometric and UHPLC-PDA

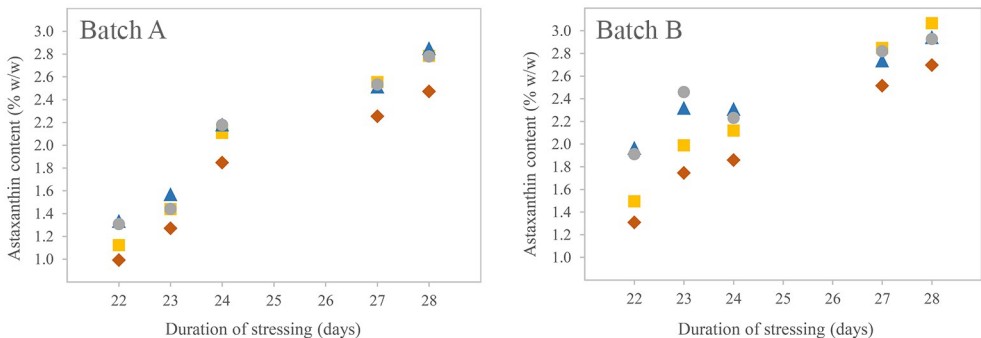

**Fig 9. Astaxanthin content in two different batches of _H. pluvialis_ cultures towards the end of a cultivation period.** Left batch A, right batch B. All-_E_-astaxanthin calculated from UHPLC-PDA measurements (◆), total astaxanthin calculated from UHPLC-PDA measurements (■), total astaxanthin calculated from a photometric approach using Eq 1 (▲) and Eq 2 (●).

approaches resulted in similar or slightly smaller astaxanthin concentrations calculated from photometric data.

The initial deviation may be due to a misestimation of other carotenoids and chlorophylls by the photometric approach. It has been shown that carotenoid composition changes during cultivation; astaxanthin increases faster than other carotenoids, and lutein and chlorophylls even decrease [44, 68, 125]. This was confirmed for lutein, as concentration fell from 0.09 to 0.06% w/w and 0.13 to 0.08% w/w in batch A and B, respectively, during the trial period. In addition, the wavelength scans of the whole extracts revealed the presence of chlorophyll _a_ by maxima at 661 and 662 nm as well as a shoulder in the carotenoid peak at 434 nm [100], which decreased over the measurement period. The extract spectra also revealed a shifting absorption maximum from 466 to 472 nm and 468 to 478 nm in batch A and B, respectively. This $\lambda_{max}$ shift was probably also due to a reduction of chlorophylls and possibly other carotenoids that absorb at shorter wavelengths while astaxanthin levels increased. This experiment was made towards the end of two cultivation periods and is in accordance with measurements of Boussiba et al. They measured absorbance spectra of extracts of _H. pluvialis_ at different cell cycle stages but observed a much greater variance when comparing green to red cells [75]. Lichtenthaler et al. especially considered and corrected for the chlorophylls in their equations, but they are based on the analysis of various plants with another carotenoid composition and without astaxanthin. Both photometric approaches assume that astaxanthin accounts for about 80% of all carotenoids, which is not true for various stress stages of _H. pluvialis_ cells, and an absorption coefficient $A_{1cm}^{1\%}$ of 1980 [100mL/(g*cm)] for the total xanthophylls [100] was assumed. Apart from the mentioned unsuitability for precise measurement of astaxanthin, it might result in a further misestimation as it also does not distinguish between the diastereomers. UHPLC-PDA analysis showed an increase in all diastereomers over the measurement periods. The relative proportion of 9_Z_- and 13_Z_-astaxanthin increased slightly initially, whereas the proportion of the di-_Z_-isomers decreased constantly. Gong et al. observed similar correlations and reported a steady increase of total astaxanthin and its isomers during cultivation but no significant difference in the relative proportion of 9_Z_- and 13_Z_-astaxanthin during the astaxanthin accumulation phase [125]. Both photometric approaches include the geometric isomers. However, their different proportions and absorption coefficients are neglected. Maximum extinction is generally reduced and shifted to shorter wavelengths in _Z_-isomers of carotenoids [54, 79]. Therefore, the exact determination of astaxanthin, including the absorption characteristics of its isomers, in complex matrices, with other carotenoids and chromophores

of unknown portions and extinction properties is a delicate task. Recent methods circumvent these problems by measuring at higher wavelengths (530 nm in DMSO), where the absorption of most carotenoids, except for astaxanthin, is near zero [48, 59, 77]. Minor deviations might still occur because of the named issues with geometrical isomers. Another possibility is to use Gauss peak spectra methods to estimate carotenoids and astaxanthin [140, 141]. The different isomer spectra can be taken into account here.

Photometric methods provided a reasonably good estimate of the total astaxanthin content at a given time point near the end of a stress period. Only minor deviations between 0.2 and 4.0% from the UHPLC-PDA-based measurements occurred. However, comparison with samples taken at an earlier stage of cultivation resulted in deviations up to 30%, with the potential for even greater discrepancy. Without further insights into the cell composition, such methods should only be used to estimate differences in astaxanthin content of the same cultures. Astaxanthin in differently cultivated *H. pluvialis* algae, strains, or even processed samples should not be compared because photometric measurements cannot specifically distinguish between isomers, other carotenoids, and chlorophylls.

## Shelf life of astaxanthin containing samples

Astaxanthin-rich biomass is often stored prior to extraction and quantification. Generally, astaxanthin is highly reactive. It reacts with oxygen and forms various auto-oxidation products with shifted absorption maxima [142] and different absorption coefficients, leading to lower extinction. Moreover, isomerization reactions of carotenoids are induced by incubation at elevated temperatures also in the absence of solvents [106]. Although this effect generally depends on the height of the temperature [58, 67, 102, 104, 105, 107], longer storage at lower temperatures may result in similar isomerization effects. To achieve reproducible results, astaxanthin losses should be minimized during storage. Therefore, the stability of free astaxanthin standard in acetone was examined at various temperatures to characterize its vulnerability without further protection. Furthermore, dried and non-dried *H. pluvialis* biomass was investigated at different temperatures and in the presence and absence of oxygen to find storage conditions that reduce astaxanthin losses and storage costs.

**Free astaxanthin standard.** Free all-*E*-astaxanthin standard with a concentration of 1 μg/mL in acetone was stored at -80˚C, -20˚C, 4˚C, and room temperature for 833 days in the dark. At room temperature, a decrease of 6% and 12% after 10 days and 41 days was observed, respectively. The concentration of samples stored in cooler conditions remained constant during this period. After 833 days, the sample stored at room temperature had lost 93% of its initial concentration, and the sample stored at 4˚C had lost 27%. For storage at -80˚C and -20˚C no major changes in optical density were observed. These results indicate partial prevention of astaxanthin degradation by temperature reduction. This assumption is supported by other studies in which increased degradation of astaxanthin was observed at elevated temperatures [76, 143, 144].

**Non-disrupted *H. pluvialis* biomass concentrated in residual medium.** *H. pluvialis* biomass suspended in a nutrient depleted liquid medium with a concentration of 40 g/L was stored at 4˚C in the dark. The measured all-*E*-astaxanthin content decreased to 93 and 83% of its initial value of 0.99±0.01% w/w (n = 6) when the samples were measured after 104 and 489 days, respectively. Similar protection of astaxanthin in frozen cells of *H. pluvialis* after 672 days was observed by Miao et al., who reported a loss of less than 15% astaxanthin [76]. *H. pluvialis* aplanospores have a rigid, multilayered cell wall that protects the alga from unfavorable environmental conditions [145, 146], which might impair oxygen diffusion into the cell and thus astaxanthin oxidation.

**Lyophilized and non-disrupted *H. pluvialis* biomass exposed to ambient atmosphere.** An aliquot of the same biomass was lyophilized and stored under ambient air atmosphere at 4°C. The all-*E*-astaxanthin content was 0.95±0.01% w/w (n = 3) directly after freeze-drying, but it decreased to 69% and 32% of this initial value when the samples were measured 108 and 489 days later, respectively. These findings are similar to those of Ahmed et al., who reported 35% astaxanthin degradation in lyophilized *H. pluvialis* after 20 weeks in non-vacuum conditions [144]. Again, the relatively high temperature and direct oxygen exposure probably promoted astaxanthin degradation. The protective sheath of the cell wall might have failed due to its desiccation compared to the previous experiments.

**Lyophilized and non-disrupted *H. pluvialis* biomass partially exposed to ambient atmosphere.** Aliquots of the same lyophilized samples as before were vacuum-sealed directly after freeze-drying. They were exposed to ambient air immediately, 7, 108, and 489 days later. These samples were measured altogether on day 489. The all-*E*-astaxanthin recoveries were 32, 44, 70, and 83% of the initial value, respectively (Fig 10 and S1 Table). The di-*Z*-isomers were generally most abundant, followed by 9*Z*- and 13*Z*-astaxanthin. Their total amount decreased the longer the sample was exposed to ambient air. However, their proportion in relation to the total astaxanthin content was similar. Exposure to oxygen has been reported to negatively affect astaxanthin during storage of *H. pluvialis* [76, 143, 144], e.g., Raposo et al. observed an improvement in astaxanthin degradation in spray-dried samples when stored under nitrogen or vacuum atmosphere compared to storage under air [71]. The small losses still observed in this experiment might be due to reactions with residual oxygen in the cells or the atmosphere of the packaging. Lyophilization was performed at 37 Pa, and no atmosphere change was applied.

**Dried and disrupted *H. pluvialis* biomass sealed and exposed to ambient air.** Two sealed bags of *H. pluvialis* powder were stored at -21°C in the dark. One of them was opened, exposed to ambient air, and closed tightly again without changing the atmosphere. The astaxanthin content of both samples was measured after two and a half years. The all-*E*-astaxanthin content was 3.98±0.04% w/w (n = 4) and 3.46±0.04% w/w (n = 5) in the sealed and the opened sample, respectively (S1 Table). Compared to the closed sample, 9*Z*- and 13*Z*-astaxanthin

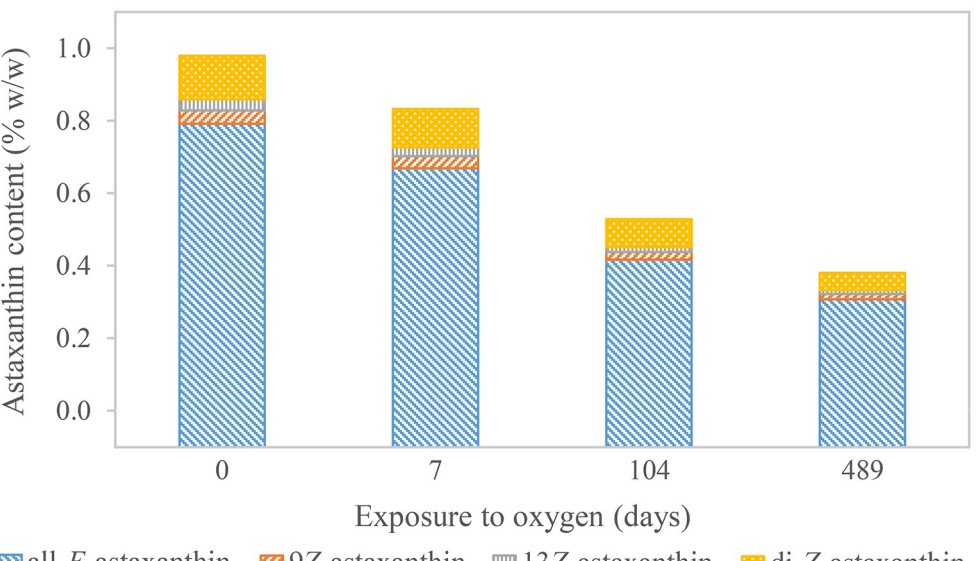

**Fig 10. Astaxanthin content in lyophilized *H. pluvialis* biomass.** It was exposed to ambient atmosphere for different periods of time and measured after 489 days.

decreased by 12% and 22%, respectively, in the opened sample, while the sum of the di-*Z*-isomers increased by 14%. Related to total astaxanthin, all-*E*-, 9*Z*-, and 13*Z*-astaxanthin were not significantly different, and only the proportion of the di-*Z*-isomers increased from 5.9±0.1 (n = 4) to 7.7±0.3% (n = 5). As described above, oxidation of astaxanthin is probably the reason for its general decrease in the opened sample. Compared to the previously analyzed lyophilized samples stored without a protective atmosphere at 4˚C, a beneficial effect of the reduced temperature was observed, as total astaxanthin content decreased less.

## Conclusion

A method for astaxanthin quantification was developed to accurately determine astaxanthin from a variety of *H. pluvialis* biomass, extracts, and formulations. Specifically, the following method parameters considered: Enzymolysis, extraction, and extract processing. Besides all-*E*-astaxanthin, the diastereomers 9*Z*-, 13*Z*- and two di-*Z*-isomers of astaxanthin were detected. In natural samples, the measurement precision of all-*E*-astaxanthin was determined with a maximum coefficient of variation of 1.1%, whereas it was below 10% regarding the diastereomers. Generally, linear correlations of biomass to astaxanthin content were determined in the extraction phase between 1.8 μg/mL and up to 30 μg/mL all-*E*-astaxanthin. It was demonstrated that an optimal concentration for quantification depended on the sample type and composition. Optimal cholesterol esterase concentration was dependent on astaxanthin concentration and biomass composition, but 2.0 units were generally sufficient in the outlined quantification range. The robustness of the method was demonstrated for ethanolic extracts of *H. pluvialis* obtained from SC-$CO_2$ extraction. Direct quantification from liquid-liquid extracts was corrected for volume aberrations and dilatations during solvent mixing.

Based on our research, we recommend starting with between 0.5 and 2.0 mg of astaxanthin-containing biomass. The initial experiment can be performed with 2.0 units of cholesterol esterase and an incubation time of 0.75 hours. The settings of the enzymolysis can be adapted depending on the type of the sample, i.e., fresh or dried biomass or extracts. For *H. pluvialis* samples, the stress level and thus estimated astaxanthin content must be considered. Different sample amounts should be processed to examine the approximate astaxanthin concentration and the influence of cellular components and to assure compliance with extraction limits. Enzyme amount and concentration can be adapted in a second measurement set if necessary. Special attention has to be paid to the correct calibration and quantification.

## Supporting information

**S1 Table. Overview of the astaxanthin content in the various experiments.**
(PDF)

**S2 Table. Overview of the astaxanthin content determined in ethanolic SC-$CO_2$ extracts.**
(PDF)

## Acknowledgments

This study was supported by Sea & Sun Technology GmbH, especially Dr. Stefan Hindersin and Clemens Elle, who provided various batches of differently processed *H. pluvialis* biomass.

## Author Contributions

**Conceptualization:** Inga K. Koopmann, Annemarie Kramer, Antje Labes.

**Data curation:** Inga K. Koopmann.

**Formal analysis:** Inga K. Koopmann.

**Funding acquisition:** Antje Labes.

**Investigation:** Inga K. Koopmann.

**Methodology:** Inga K. Koopmann, Annemarie Kramer.

**Project administration:** Annemarie Kramer.

**Supervision:** Annemarie Kramer, Antje Labes.

**Validation:** Inga K. Koopmann.

**Visualization:** Inga K. Koopmann.

**Writing – original draft:** Inga K. Koopmann.

**Writing – review & editing:** Inga K. Koopmann, Annemarie Kramer, Antje Labes.

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
