## [Decision Letter · Decision Letter 0]

2 May 2022

PONE-D-22-07202Development and validation of reliable astaxanthin quantification from natural sourcesPLOS ONE

Dear Dr. Antje Labes,

Thank you for submitting your manuscript to PLOS ONE. After careful consideration, we feel that it has merit but does not fully meet PLOS ONE’s publication criteria as it currently stands. Therefore, we invite you to submit a revised version of the manuscript that addresses the points raised during the review process.

We look forward to receiving your revised manuscript.

Kind regards,

Vandana Vinayak, PhD

Academic Editor

PLOS ONE

Journal Requirements:

"IK has worked for Sea and Sun Technology, a provider of H. pluvialis biomass for the study. All authors have declared that no other competing interests exist."

Additional Editor Comments:

Dear Dr. Antje Labes,

I can now inform you that the reviewers and editor have evaluated the manuscript PONE-D-22-07202

Development and validation of reliable astaxanthin quantification from natural sources.

Please consider the reviews to see if revision would be feasible. Should you wish to resubmit you should explain how and where each point of the reviewers' comments has been incorporated. For this, use submission item "Revision Notes" when uploading your revision. Also, indicate the changes in marked changes version of the revised manuscript (submission item "Revision, changes marked").

Regards

Dr. Vandana Vinayak

Academic Editor

Reviewers' comments:

Reviewer's Responses to Questions

**Comments to the Author**

1. Is the manuscript technically sound, and do the data support the conclusions?

Reviewer #1: Partly

Reviewer #2: Yes

Reviewer #3: Yes

Reviewer #4: Yes

2. Has the statistical analysis been performed appropriately and rigorously? 

Reviewer #1: No

Reviewer #2: Yes

Reviewer #3: Yes

Reviewer #4: Yes

3. Have the authors made all data underlying the findings in their manuscript fully available?

Reviewer #1: Yes

Reviewer #2: Yes

Reviewer #3: Yes

Reviewer #4: No

4. Is the manuscript presented in an intelligible fashion and written in standard English?

Reviewer #1: No

Reviewer #2: Yes

Reviewer #3: Yes

Reviewer #4: Yes

5. Review Comments to the Author

Reviewer #1: Comments:

Sentence 41-43: “calibration……5%” seems incomplete and complicated statement.

Sentence 44: Rewrite it, as it seems the two di-Z forms are 9Z and 13Z mentioned in the sentence and if so why to mention the “two di-Z forms” in the same sentence again?

Sentence 47-49: only the dried biomass was processed for Astaxanthin and not the liquid culture or the fresh biomass?

Sentence 49-50: Cholesterol esterase (1.0 to 2.0 U) is per how many mg or mL of dried biomass or culture respectively?

Sentence 50-52: This line is not necessary here; it should be discussed in introduction section.

Sentence 53: “The reliability of photometric Astaxanthin estimation was assessed”, how? Name the method or explain in one sentence.

It’s not clear in the abstract if the authors are comparing two methods of extraction or estimation or both, please clarify.

Sentence 61-63: needs reference.

Sentence 68-69: what about the allergenicity of Astaxanthin if provided as food supplement.

Sentence 84-89: it gives explanation of why this technique is needed. The limitations due to which related techniques like this have not been used should also be stated.

Sentence 100: which wavelength?

Sentence 119-128: it gives the glimpse of steps involved, but in the introduction it was claimed that it is a shorter and faster method. However, Astaxanthin extraction, enzymolysis, liquid-liquid extraction and then analysis with UHPLC,UV/VIS spectrophotometry does not seem to be a short method.

Sentence 156: what was the make of beat miller, diameter of zirconium beads, time of beat milling, and the cooling time in the process?

Sentence 167: is the fresh biomass dewatered?

Sentence 172: Mention 3.3U/mL as cholesterol esterase concentration in abstract used for de-esterification.

Sentence 211: how the proportion of 69.4% was assumed?

Sentence 215: replace solved to dissolve.

Sentence 223: is 45 min. (0.75 h) is the minimum time used, why authors did not try the lower incubation period or higher incubation period than to 1.5 h.

Sentence 260-265: previously, it was mentioned that additional incubation time of 1.5 h was also used so; for detection limit and linearity, why only one incubation time is used?

Sentence 267-271: there is lot of confusion between the samples (fresh or dry), if fresh then how it was weighed and what is the incubation period in no. of days decided on what basis, which two samples were chosen and why?

I cannot follow the series of experiments performed and what samples were used what was the incubation period, was there any stress? I suggest authors to prepare an experimental design section with text and detail figure to explain precisely how the experiments are performed with each step and from which step sample were taken for which analysis and why?

Sentence 287: On what basis it was considered 4.5 % w/w of the H. pluvialis is Astaxanthin?

Sentence 300: In sentence 270-271, four days are mentioned in sentence 300 five days are mentioned, why? It’s confusing to follow.

In figure 5: It seems like there is no effect of increasing the biomass (mg) but, how it is possible if considered 4.5 % w/w of the H. pluvialis is Astaxanthin?

In figure 7: give legends with in the graph for samples like M1, M2 ETC.

In figure 9: which stress is imposed and why? Also indicate level of significance in each figure.

Conclusion is too long; make it precise and conclusive in 10-12 sentences. For instance sentence 986 seems like repetition of first line of conclusion. Comparison should be a part of discussion. Similarly sentence 990-991 should be discussed in methodology section.

Reviewer #2: Comments:

1. In line no 44-45 sentence “In extracts from H. pluvialis, the observed measurement range was extended to 30 µg/mL”. What was the amount of H. pluvialis biomass taken? It should be mentioned in the abstract also.

2. Line no 47, 48, 48 ‘The precision of all-E-astaxanthin quantification in dried H. pluvialis biomass was calculated with a coefficient of variation of maximal 1.1%, whereas it was below 10% regarding the diastereomers” What was the amount of H. pluvialis biomass?

3. In the beginning sentence of introduction “Astaxanthin (3,3´-dihydroxy-β,β´-carotene-4,4´-dione) is a secondary ketocarotenoid. It has a hydrocarbon backbone that comprises a central, delocalized π-electron system. β-ionone rings terminate the hydrocarbon chain at both ends. The presence of one hydroxy- and one oxo-group at each of these terminal rings further classify it as xanthophyll.” Add reference and for more information prefer the suggested article “Light modulates transcriptomic dynamics upregulating astaxanthin accumulation in Haematococcus: A review” https://doi.org/10.1016/j.biortech.2021.125707

4. Add new references from line no 84-89.

5. Line no 157, what was the percentage of acetone used for astaxanthin extraction? Likewise line no 174 mention the percentage of petroleum ether?

6. Line no 161 please check whether it is 10.000 x g or 10,000 x g. See at other places also.

7. Mention makes and model of the instruments/ equipments used during the experiment. For instance, vortex and ultrasonicate.

8. Line no 190 and 191 ‘Optical spectra were measured in a range of 200 to 800 nm, and astaxanthin data were analyzed and quantified at 474 nm. Lutein was quantified at 448 nm.’ Add the references for astaxanthin and lutein as well, to increase the novelty of the article.

9. Line no 215 please check whether “solved” is the right word to use here. Check in the entire manuscript.

10. Check for typographical errors in the entire manuscript.

11. Measuring units should be written clearly for instance see line no 221-224.

12. Text in the conclusion should be reduced to 8-12 lines.

Reviewer #3: In the manuscript “Development and validation of reliable astaxanthin quantification from natural sources” Authors developed enzymolysis-based astaxanthin quantification method to hydrolyse astaxanthin esters and determine free astaxanthin in all its diastereomeric forms. The investigated results could be useful for understanding the scientific knowledge. Therefore, I recommend this study for the publication in PLOS ONE after answering the following queries.

• Author said that, mostly sophisticated techniques as liquid chromatography and spectrophotometric/mass spectrometry are used for identification and quantification of astaxanthin. However author themselves used ultra-high performance liquid chromatography (UHPLC) and UV/VIS spectrometry techniques in the present study; justify

• In the calibration curves of astaxanthin standards in table 1; why different number of measurements was chosen?

• For detection limits of astaxanthin determination why selected specific biomass quantity of H. pluvialis (from 0.04 to 3.98 mg), saponification time (0.75 h) and enzyme concentration (2.0 U).

• Why astaxanthin concentration deceased in the sample exposed to ambient air (3.46±0.04% w/w) as compared to vacuumed samples (3.98±0.04% w/w) after prolong storage?

• There is poor English language at many places in the manuscript, so author needs to check thoroughly and improve.

Reviewer #4: Astaxanthin is a biomolecule with a very high added value and with promising/already established applications, including in biomedicine. For at least economic and correct dosage, an accurate method for astaxanthin quantification is required. The manuscript by Koopmann and collaborators enters in this frame and therefore appears as timely and witha high potential interest. The quantification of astaxanthin as other carotenoids is difficult because astaxanthin is actually constituted by several diastereoisomers, the relative abundance of which depending on many factors, including extraction and storing conditions. In addition, depending on which phylum the source belongs, different enantiomers are found. The manuscript is well written even if I found it difficult to read in some places. Concerning language issues, I have proposed some modifications that the authors should evaluate before an eventual validation because I am not a native English writter. A detailled llist of comments is displayed below. Altogether, I found the manuscript interesting but difficult to use. For increasing its usefulness, I suggest to the authors to add (1) an additional figure displaying a logic scheme to allow the readers to choose what to do regarding quantification objectives and (2) the corresponding finalized protocols. Both could be added as supplemental data

l88: add a citation

l100: 'suggesting using another wavelength' -> 'suggesting the use of another wavelength'

l104: already indicated above

l105: 'a similar approach' but related to what?

l110 should continue l109

l213: what is the source of lutein?

l240 and throughout the manuscript: check the use of '.' in figures. Here 3.000 = 3 x g ... Centrifuge aceleration are usually written under this format XXX x g (with g in italic to avoid confusion with the mass unit).

l253 and throughout the manuscript: I suggest to use (X) rather than x) and to avoid '.' before. Actually, the (x) indicates a sucession of items.

l268: Are the two samples arising from the same batch?

l302: which sample?

l313 and throughout the manuscript: italize 'a' of chlorophyll a'

l327: give a range for the vacuum or at least how it was performed

l362: why the linear regression does not cross the (0,0) coordinate?

l390-394: indicate the meaning of each symbol both in the graph and in the legend

l396: by 11%?

l402-405: at least a citation about this is required. Pictures comparing the phases would also help

l405-407: Is there any reason for this different behaviour?

l412-418: it would help the reader to reproduce the separation if the text would be accompanied by a table providing (1) log k', (2) the wavelength maximum/maxima in the eluting solvent and (3) in a solvent of reference

l421: 'dissolved' instead of 'solved'?

l426: would be better to indicate it as relative amount

l431: could the author observe the cis-peak?

l442: I do not understand 'these significant differences'

Fig3

- why are m/z and UV/vis bandwidths different?

- the resolution and the contrast of the magnified parts is very weak. Please increase

- could other keto-carotenoid be detected?

l560: 'small': really, Is not reaching up to 10%? In addition, small is a very relative term. Better to give a range in percent of the volume. No emulsion at the interface of the two phases?

l639-640: I do not understand clearly if it is the same sample or different samples.

l640: '3 and 4 different days' of what?

l659: I do not understand towhat correspond the 'differentlly cultivated'

Fig6 and the next figures: figure out the 100%, for instance by a dashed line. How the 100% was determined?

l680: 'different conditions': specify them

l684: indicate if the biomass is in DW or FW

l720: remove 'also'

l722-728: could this part summarized by writting that the optimal concentration ranges between 10 et 18%?

l725: 'other enzymes': could the authors be more preciseN

l729: '... without ethanol': this study? If not, add a citation

Fig8

- I could not find the call to this figure

- what is the meaning of (-) in the title of the X axis?

l753! 'maximum': how was it measured?

l754: ... mg oleoresin': DW or FW?

l761: replace 'exceeding' by 'an excess'?

l762: about carotenoids! astaxanthin belongs to carotenoid?

l763: why competition? Could the authors elaborate a bity on this?

l764: indicate the dilution factor

l774-775: I do not understand this sentence

l783: 'alga' instead of 'algae'

l783-784! I would replace 'and' by 'or'

l791! 'higher proportion' in O3 but not in O1 and O2. Are the fdifference significant?

l797-799: indicate the number of repetition

l808: I do not understand

l811-812: and what?

l828-830: this is strange, is not it? Please elaborate on this.

l843: 'general equation': which one?

l845-851: no statistics?

l853-856: write 'Batch A' and 'Batch B' on figure 9

l863: 616 and 662 nm are not characteristic of chloorphyll a and chlorophyll b, respectively

l870-878: rather obvious. Could this part reduced?

l916: give the solvent in which E-all-astaxanthin has been dissolved

l929: italize 'H.' in 'H. pluvialis'

l936

- of course, it cannot be the same biomass? Was it an aliquot?

- I would replace 'it' by 'the'

l953: e.g.

l965: under which type of atmosphere?

l968-973: rather complicated section

l969-970! 12% and 22% regarding what?

Missing citations

Kopecky, J., et al. (2000). "Microalgae as a source for secondary carotenoid production: a screening study." Algological Studies 98: 153-168

Schoefs, B., et al. (2001). "Astaxanthin accumulation in Haematococcus requires a cytochrome P450 hydroxylase and an active synthesis of fatty acids." FEBS LETTERS 500(3): 125-128.

Lemoine, Y. and B. Schoefs (2010). "Secondary ketocarotenoid astaxanthin biosynthesis in algae: a multifunctional response to stress." Photosynthesis Research 106: 155-177

Gateau, H., et al. (2017). "Carotenoids of microalgae used in food industry and medicine." Mini-Review in Medicinal Chemistry 17: 1140-1172

Solymosi, K., et al. (2015). Food colour additives of natural origin. Colour Additives for Foods and Beverages: Development, Safety and Applications. M. Scotter, Woodhead Publishing: 1-34.

Scarsini, M., et al. (2020). Carotenoid overproduction in microalgae: Biochemical and genetic engineering. Pigments from Microalgae Handbook. E. Jacob-Lopes, M. I. Queiroz and L. Q. Zepka. Cham, Springer International Publishing: 81-126

Schoefs, B. (2003). "Chlorophyll and carotenoid analysis in food products. A practical case-by-case view." Trends in Analytical Chemistry 22(6): 335-339.

6. PLOS authors have the option to publish the peer review history of their article (what does this mean?). If published, this will include your full peer review and any attached files.

Reviewer #1: No

Reviewer #2: No

Reviewer #3: No

Reviewer #4: No

---

## [Author Response · Author response to Decision Letter 0]

5 Aug 2022

'Response to Reviewers' for the manuscript: “Development and validation of reliable astaxanthin quantification from natural sources”. PONE-D-22-07202

Reviewer #1: 

1. Sentence 41-43: “calibration……5%” seems incomplete and complicated statement. 

We adapted the sentence. 

2. Sentence 44: Rewrite it, as it seems the two di-Z forms are 9Z and 13Z mentioned in the sentence and if so why to mention the “two di-Z forms” in the same sentence again? 

We adapted the sentence.

3. Sentence 47-49: only the dried biomass was processed for Astaxanthin and not the liquid culture or the fresh biomass? 

Yes, the precision of the method was determined by using only dried H. pluvialis biomass.

4. Sentence 49-50: Cholesterol esterase (1.0 to 2.0 U) is per how many mg or mL of dried biomass or culture respectively? 

We added the maximum biomass used in these experiments.

5. Sentence 50-52: This line is not necessary here; it should be discussed in introduction section. 

We kept the sentence because it helps the reader already on abstract level to see what conclusions can be found in the article.

6. Sentence 53: “The reliability of photometric Astaxanthin estimation was assessed”, how? Name the method or explain in one sentence. 

It was assessed by comparing with the developed method. We added the information.

7. It’s not clear in the abstract if the authors are comparing two methods of extraction or estimation or both, please clarify. By clarifying the above mentioned problem, this is hopefully also explained.

8. Sentence 61-63: needs reference. We added references.

9. Sentence 68-69: what about the allergenicity of Astaxanthin if provided as food supplement. 

No allergic reactions to astaxanthin have been described in the literature. We added the information.

10. Sentence 84-89: it gives explanation of why this technique is needed. The limitations due to which related techniques like this have not been used should also be stated. 

We have worked out the problem. 

11. Sentence 100: which wavelength? 

We added the information.

12. Sentence 119-128: it gives the glimpse of steps involved, but in the introduction it was claimed that it is a shorter and faster method. However, Astaxanthin extraction, enzymolysis, liquid-liquid extraction and then analysis with UHPLC,UV/VIS spectrophotometry does not seem to be a short method. 

We adapted the sentence for better understanding.

13. Sentence 156: what was the make of beat miller, diameter of zirconium beads, time of beat milling, and the cooling time in the process? 

We added the bead specifications. The make of the bead miller is described in line 160-161 (original manuscript). The milling time is described in line 161 (original manuscript). There was no cooling performed. We added the information.

14. Sentence 167: is the fresh biomass dewatered? 

No, it was not. We added a word for better understanding.

15. Sentence 172: Mention 3.3U/mL as cholesterol esterase concentration in abstract used for de-esterification. 

We added a further sentence for better understanding and added the information of applied Units per biomass in the abstract.

16. Sentence 211: how the proportion of 69.4% was assumed? 

By calculating the molecular weight. We added the information.

17. Sentence 215: replace solved to dissolve. 

We changed the wording.

18. Sentence 223: is 45 min. (0.75 h) is the minimum time used, why authors did not try the lower incubation period or higher incubation period than to 1.5 h. 

For the choice of the incubation time we used available protocols (references given in line 119 original manuscript). Longer incubation was not useful for us because 1.5 h did not show any advantages over 0.75 h.

19. Sentence 260-265: previously, it was mentioned that additional incubation time of 1.5 h was also used so; for detection limit and linearity, why only one incubation time is used? 

This is used, because it had the best results and was used for all other experiments. There is no statistic difference in all-E-astaxanthin between an incubation time of 0.75 and 1.5 h. We did not see the necessity to determine these parameters for an inferior method. 

20. Sentence 267-271: there is lot of confusion between the samples (fresh or dry), if fresh then how it was weighed and what is the incubation period in no. of days decided on what basis, which two samples were chosen and why? 

The samples were dried (line 268, original manuscript). We bought theses samples as explained in line 143 (original manuscripts). On all the samples mentioned so far, we had no influence and we did not get any detailed information about the cultivation parameters or harvest unless of knowing that they comprised stressed cells. Samples from Sea & Sun Technology GmbH were stressed by natural light. Only for two batches, which were used for the photometric experiments, we got some information and we changed the paragraph “H. pluvialis biomass and astaxanthin containing extracts” (starting at line 141 original manuscript) for better understanding.

21. I cannot follow the series of experiments performed and what samples were used what was the incubation period, was there any stress? I suggest authors to prepare an experimental design section with text and detail figure to explain precisely how the experiments are performed with each step and from which step sample were taken for which analysis and why? 

That is why we have figure 1. We made a stronger connection between manuscript and the figure by adding letters to the boxes that can also be found in the text.

22. Sentence 287: On what basis it was considered 4.5 % w/w of the H. pluvialis is Astaxanthin?

By measurement. We adapted a series of sentences for better understanding.

23. Sentence 300: In sentence 270-271, four days are mentioned in sentence 300 five days are mentioned, why? It’s confusing to follow. 

These are completely independent experiments. For clarification, we adapted the sentence and added a link to figure 1.

24. In figure 5: It seems like there is no effect of increasing the biomass (mg) but, how it is possible if considered 4.5 % w/w of the H. pluvialis is Astaxanthin? 

Figure 5 illustrates the relationship between biomass inputs and the respective relative astaxanthin content in percent per biomass. There should be no effect of increasing biomass until the point of overloading of the enzyme, which is somewhere above 2.0 mg. This figure demonstrates that the method gets equal results over a broad range of different biomass inputs. We added this to the figure title. 

25. In figure 7: give legends with in the graph for samples like M1, M2 ETC. 

We added the information about the missing information in materials and methods and adapted the figure title for better understanding.

26. In figure 9: which stress is imposed and why? Also indicate level of significance in each figure.

We added all that we knew about the stress conditions in the text above. Significance was not determined. Note that this experiment itself was performed in a replicate of two batches for five days.

27. Conclusion is too long; make it precise and conclusive in 10-12 sentences. For instance sentence 986 seems like repetition of first line of conclusion. Comparison should be a part of discussion. Similarly sentence 990-991 should be discussed in methodology section. 

We shortened the conclusion and made it more concise. We deleted the sentence in line 986 (original manuscript). We kept he sentence in line 990-991 (original manuscript) as it is important to us that the method is able to determine astaxanthin from ethanolic extracts, which are very common in the industrial production of astaxanthin-rich products, without further processing. We left an additional part for recommending the reader how to start measuring samples with unknown astaxanthin concentration.

Reviewer #2

1. In line no 44-45 sentence “In extracts from H. pluvialis, the observed measurement range was extended to 30 µg/mL”. What was the amount of H. pluvialis biomass taken? It should be mentioned in the abstract also. 

We adapted the sentence. The exact specification of biomass is not useful in this case, because biological samples have a high variability and a blanket statement would be misleading.

2. Line no 47, 48, 48 ‘The precision of all-E-astaxanthin quantification in dried H. pluvialis biomass was calculated with a coefficient of variation of maximal 1.1%, whereas it was below 10% regarding the diastereomers” What was the amount of H. pluvialis biomass? 

We added the information. 

3. In the beginning sentence of introduction “Astaxanthin (3,3´-dihydroxy-β,β´-carotene-4,4´-dione) is a secondary ketocarotenoid. It has a hydrocarbon backbone that comprises a central, delocalized π-electron system. β-ionone rings terminate the hydrocarbon chain at both ends. The presence of one hydroxy- and one oxo-group at each of these terminal rings further classify it as xanthophyll.” Add reference and for more information prefer the suggested article “Light modulates transcriptomic dynamics upregulating astaxanthin accumulation in Haematococcus: A review” https://doi.org/10.1016/j.biortech.2021.125707

We added references.

4. Add new references from line no 84-89. 

We added references.

5. Line no 157, what was the percentage of acetone used for astaxanthin extraction? Likewise line no 174 mention the percentage of petroleum ether? 

We added the percentages of acetone and petroleum ether in the paragraph beginning at line 156 (original manuscript). 

6. Line no 161 please check whether it is 10.000 x g or 10,000 x g. See at other places also. It is 10,000 x g. 

We corrected the mistake.

7. Mention makes and model of the instruments/ equipments used during the experiment. For instance, vortex and ultrasonicate. 

We added the information.

8. Line no 190 and 191 ‘Optical spectra were measured in a range of 200 to 800 nm, and astaxanthin data were analyzed and quantified at 474 nm. Lutein was quantified at 448 nm.’ Add the references for astaxanthin and lutein as well, to increase the novelty of the article. 

We added the needed information.

9. Line no 215 please check whether “solved” is the right word to use here. Check in the entire manuscript. 

It was not. We corrected the mistake.

10. Check for typographical errors in the entire manuscript. 

We did.

11. Measuring units should be written clearly for instance see line no 221-224. 

We changed the measuring units in the entire manuscript.

12. Text in the conclusion should be reduced to 8-12 lines.

We shortened the conclusion and made it more concise. We left an additional part for recommending the reader how to start measuring samples with unknown astaxanthin concentration.

Reviewer #3:

1. Author said that, mostly sophisticated techniques as liquid chromatography and spectrophotometric/mass spectrometry are used for identification and quantification of astaxanthin. However author themselves used ultra-high performance liquid chromatography (UHPLC) and UV/VIS spectrometry techniques in the present study; justify. We used UHPLC (and HPLC is possibly similar) but without the need to detect and quantify all the various esters of astaxanthin.

We added a sentence for further clarification.

2. In the calibration curves of astaxanthin standards in table 1; why different number of measurements was chosen? Measurement of all-E-astaxanthin has a greater number of measurements because it was used as parallel check for device measurement stability. 

We added a statement for clarification.

3. For detection limits of astaxanthin determination why selected specific biomass quantity of H. pluvialis (from 0.04 to 3.98 mg), saponification time (0.75 h) and enzyme concentration (2.0 U). 

or the choice of the incubation time we used protocols (references given in line 119 original manuscript and we added a reference). Longer incubation was not useful for us because 1.5 h did not show any advantages over 0.75 h.

4. Why astaxanthin concentration deceased in the sample exposed to ambient air (3.46±0.04% w/w) as compared to vacuumed samples (3.98±0.04% w/w) after prolong storage? 

We added a sentence and adapted the paragraph for better explanation.

5. There is poor English language at many places in the manuscript, so author needs to check thoroughly and improve. 

We did a thoughout revision of the whole manuscript.

Reviewer #4:

1. For increasing its usefulness, I suggest to the authors to add (1) an additional figure displaying a logic scheme to allow the readers to choose what to do regarding quantification objectives and (2) the corresponding finalized protocols. Both could be added as supplemental data. (1)

That is why we have figure 1. We made a stronger connection between manuscript and the figure by adding letters to the boxes that can also be found in the text. (2) Finalized protocols might be misleading due to the variability of astaxanthin in natural samples. Instead, we gave a recommendation of how to approach the measurement when not knowing the astaxanthin levels in the conclusion section. 

2. l88: add a citation 

We did.

3. l100: 'suggesting using another wavelength' -> 'suggesting the use of another wavelength'

we adapted the sentence accordingly.

4. l104: already indicated above 

We left the sentence as it is.

5. l105: 'a similar approach' but related to what? 

We adapted the sentence for better understanding.

6. l110 should continue l109 

We changed the sentence to have a stringent logic structure of the paragraphs. Here a new paragraph is starting.

7. l213: what is the source of lutein? Also H. pluvialis. 

We added the information.

8. l240 and throughout the manuscript: check the use of '.' in figures. Here 3.000 = 3 x g ... Centrifuge aceleration are usually written under this format XXX x g (with g in italic to avoid confusion with the mass unit).

We corrected the delimiters and wrote the “g” in italics.

9. l253 and throughout the manuscript: I suggest to use (X) rather than x) and to avoid '.' before. Actually, the (x) indicates a sucession of items. 

We used double parenthesis. We decided to leave the dot for increasing readability especially in long enumerations.

10. l268: Are the two samples arising from the same batch? 

No, we added the information.

11. l302: which sample?

We adapted the sentence and added a further one for better understanding.

12. l313 and throughout the manuscript: italize 'a' of chlorophyll a' 

We did.

13. l327: give a range for the vacuum or at least how it was performed

We did.

14. l362: why the linear regression does not cross the (0,0) coordinate? 

This is an observation we made when calculating the calibration curves. Many experiments were performed to get the curve through 0/0, but it did not work. It might be a problem with the measurement device (high noise levels). 

15. l390-394: indicate the meaning of each symbol both in the graph and in the legend 

All meanings are explained in the legend. For a clear and distinguished data representation in the figure, we decided against a legend in the figure. 

16. l396: by 11%? 

Yes. We corrected that.

17. l402-405: at least a citation about this is required. Pictures comparing the phases would also help 

We wanted to cite this, but the only references we found were about the solubility in other solvents (DCM, TCM, DMSO, methanol, acetone), so we relied on our own experiments. We decided against a picture.

18. l405-407: Is there any reason for this different behaviour? 

We explained in the following lines.

19. l412-418: it would help the reader to reproduce the separation if the text would be accompanied by a table providing (1) log k', (2) the wavelength maximum/maxima in the eluting solvent and (3) in a solvent of reference 

For the wavelength maxima and retention times we have got figure 3. We had no standards for the various Z-isomers, so we could not record their individual properties in other solvents.

20. l421: 'dissolved' instead of 'solved'? 

Yes. We corrected the mistake.

21. l426: would be better to indicate it as relative amount 

Yes. We indicated the relative amount.

22. l431: could the author observe the cis-peak? 

Yes, see line 431 (original manuscript).

23. l442: I do not understand 'these significant differences' 

We clarified the sentence.

24. Fig3: why are m/z and UV/vis bandwidths different? 

As processed by the software. We added the signal strength to the chromatogram.

25. Fig. 3 - the resolution and the contrast of the magnified parts is very weak. Please increase 

In order to show all pictures together, we have decided for this representation. All relevant data has been added as text.

26. - could other keto-carotenoid be detected? 

Not in this sample. Lutein could be detected but was not of interest for this study. 

27. l560: 'small': really, Is not reaching up to 10%? In addition, small is a very relative term. Better to give a range in percent of the volume. No emulsion at the interface of the two phases?

We deleted the word “small”. We did not want to give a clear percentage here, because it could also be water that is transferred. There was no emulsion, only a very clear phase boundary. 

28. l639-640: I do not understand clearly if it is the same sample or different samples. 

We clarified

29. l640: '3 and 4 different days' of what? 

We clarified

30. l659: I do not understand towhat correspond the 'differentlly cultivated' 

We deleted that term and added a paragraph about the samples in the material and methods part.

31. Fig6 and the next figures: figure out the 100%, for instance by a dashed line. How the 100% was determined?

 % w/w means the weight percentage of astaxanthin per dry biomass. In samples with biological origin, 100 % cannot be reached. 

32. l680: 'different conditions': specify them 

We can not. As mentioned, we added a paragraph for further clarification. 

33. l684: indicate if the biomass is in DW or FW ´

We did.

34. l720: remove 'also' 

We did.

35. l722-728: could this part summarized by writting that the optimal concentration ranges between 10 et 18%? There is no optimal concentration range. It can only be concluded that there is no effect of ethanol on the quantification of all-E-astaxanthin between 0 and 18 % v/v. 

We think this is important and have not summarized it.

36. l725: 'other enzymes': could the authors be more preciseN A lipase from P. aeruginosa. 

We added the information.

37. l729: '... without ethanol': this study? If not, add a citation 

This study.

38. Fig8 - I could not find the call to this figure 

The call is in line 791 (original manuscript). 

39. - what is the meaning of (-) in the title of the X axis? 

We deleted all dashes.

40. l753! 'maximum': how was it measured? 

We corrected the mistake. The maximum was measured with the developed method. It is important that this is only the maximum determinable astaxanthin proportion. It cannot be excluded that there is even more astaxanthin in the sample. We adapted the text for a better understanding.

41. l754: ... mg oleoresin': DW or FW?

The oleoresin was weighed as it was (liquid). l761: replace 'exceeding' by 'an excess'? We did.

42. l762: about carotenoids! astaxanthin belongs to carotenoid? 

Yes, it does.

43. l763: why competition? Could the authors elaborate a bity on this? 

We adapted the sentence.

44. l764: indicate the dilution factor 

The information was disclosed by the provider. We deleted the misleading sentence.

45. l774-775: I do not understand this sentence 

We rephrased it.

46. l783: 'alga' instead of 'algae' 

Yes, we corrected it.

47. l783-784! I would replace 'and' by 'or' 

Yes, we changed it. 

48. l791! 'higher proportion' in O3 but not in O1 and O2. Are the fdifference significant? 

They were significantly higher in all samples. We clarified by adapting the sentence. 

49. l797-799: indicate the number of repetition 

We adapted the figure description and also the description of the preceding figure. 

50. l808: I do not understand 

We rephrased the sentence.

51. l811-812: and what? 

This should only show that our premise was met in the first place.

52. l828-830: this is strange, is not it? Please elaborate on this. 

We rephrased. 

53. l843: 'general equation': which one? 

We added the number of the equation

54. l845-851: no statistics? 

No statistics. The two batches and number of different trials are equivalent to replicates. 

55. l853-856: write 'Batch A' and 'Batch B' on figure 9 

We adapted the figure accordingly. 

56. l863: 616 and 662 nm are not characteristic of chloorphyll a and chlorophyll b, respectively 

616 was a typographical mistake. Still, we deleted the “chlorophyll b” because it was lacking evidence.

57. l870-878: rather obvious. Could this part reduced? 

It is rather obvious, still we wonder why the photometric approach is used so often in literature and its disadvantages are widely ignored. This is why we wanted to stress this here. 

58. l916: give the solvent in which E-all-astaxanthin has been dissolved 

We did. 

59. l929: italize 'H.' in 'H. pluvialis' 

We did.

60. - of course, it cannot be the same biomass? Was it an aliquot? 

Yes, we corrected the sentence.

61. - I would replace 'it' by 'the' 

We could not find an “it” but we replaced the “its”. 

62. l953: e.g. 

We changed the sentence.

63. l965: under which type of atmosphere? Air. 

We added some words for clarification.

64. l968-973: rather complicated section 

We tried to rearrange and add some statements for clarification.

65. l969-970! 12% and 22% regarding what? 

Compared to the closed sample. We clarified the sentence.

66. Missing citations

Kopecky, J., et al. (2000). "Microalgae as a source for secondary carotenoid production: a screening study." Algological Studies 98: 153-168

Schoefs, B., et al. (2001). "Astaxanthin accumulation in Haematococcus requires a cytochrome P450 hydroxylase and an active synthesis of fatty acids." FEBS LETTERS 500(3): 125-128.

Lemoine, Y. and B. Schoefs (2010). "Secondary ketocarotenoid astaxanthin biosynthesis in algae: a multifunctional response to stress." Photosynthesis Research 106: 155-177

Gateau, H., et al. (2017). "Carotenoids of microalgae used in food industry and medicine." Mini-Review in Medicinal Chemistry 17: 1140-1172

Solymosi, K., et al. (2015). Food colour additives of natural origin. Colour Additives for Foods and Beverages: Development, Safety and Applications. M. Scotter, Woodhead Publishing: 1-34.

Scarsini, M., et al. (2020). Carotenoid overproduction in microalgae: Biochemical and genetic engineering. Pigments from Microalgae Handbook. E. Jacob-Lopes, M. I. Queiroz and L. Q. Zepka. Cham, Springer International Publishing: 81-126

Schoefs, B. (2003). "Chlorophyll and carotenoid analysis in food products. A practical case-by-case view." Trends in Analytical Chemistry 22(6): 335-339. Thanks for your recommendation.

We added two of the references. Our presented data is already covered by other literature.

---

## [Decision Letter · Decision Letter 1]

18 Nov 2022

Development and validation of reliable astaxanthin quantification from natural sources

PONE-D-22-07202R1

Dear Dr. Labes,

We’re pleased to inform you that your manuscript has been judged scientifically suitable for publication and will be formally accepted for publication once it meets all outstanding technical requirements.

Kind regards,

Vandana Vinayak, PhD

Academic Editor

PLOS ONE

Additional Editor Comments (optional):

authors have revised the manuscript and may be published as per Journals policy

Reviewers' comments:

Reviewer's Responses to Questions

**Comments to the Author**

1. If the authors have adequately addressed your comments raised in a previous round of review and you feel that this manuscript is now acceptable for publication, you may indicate that here to bypass the “Comments to the Author” section, enter your conflict of interest statement in the “Confidential to Editor” section, and submit your "Accept" recommendation.

Reviewer #1: All comments have been addressed

Reviewer #2: All comments have been addressed

2. Is the manuscript technically sound, and do the data support the conclusions?

Reviewer #1: Yes

Reviewer #2: Yes

3. Has the statistical analysis been performed appropriately and rigorously? 

Reviewer #1: Yes

Reviewer #2: Yes

4. Have the authors made all data underlying the findings in their manuscript fully available?

Reviewer #1: Yes

Reviewer #2: Yes

5. Is the manuscript presented in an intelligible fashion and written in standard English?

Reviewer #1: Yes

Reviewer #2: Yes

6. Review Comments to the Author

Reviewer #1: The manuscript addresses the problems of astaxanthin quantification, and it has significant scientific data to provide answers to the problems in quantification

Reviewer #2: Authors have answered all the comments correctly. This article is recommended for the possible publication.

7. PLOS authors have the option to publish the peer review history of their article (what does this mean?). If published, this will include your full peer review and any attached files.

Reviewer #1: **Yes: **Dr. Mrinal Kashyap

Reviewer #2: No

---

## [Editor Report · Acceptance letter]

23 Nov 2022

PONE-D-22-07202R1 

Development and validation of reliable astaxanthin quantification from natural sources 

Dear Dr. Labes:

I'm pleased to inform you that your manuscript has been deemed suitable for publication in PLOS ONE. Congratulations! Your manuscript is now with our production department. 

Kind regards, 

on behalf of

Dr. Vandana Vinayak 

Academic Editor

PLOS ONE